# FairRARI: A Plug and Play Framework for Fairness-Aware PageRank

**Emmanouil Kariotakis** [1]   **Aritra Konar** [1]

## Abstract

PageRank (PR) is a fundamental algorithm in graph machine learning tasks. Owing to the increasing importance of algorithmic fairness, we consider the problem of computing PR vectors subject to various group-fairness criteria based on sensitive attributes of the vertices. At present, principled algorithms for this problem are lacking - some cannot guarantee that a target fairness level is achieved, while others do not feature optimality guarantees. In order to overcome these shortcomings, we put forth a unified in-processing convex optimization framework, termed FairRARI, for tackling different group-fairness criteria in a "plug and play" fashion. Leveraging a variational formulation of PR, the framework computes fair PR vectors by solving a strongly convex optimization problem with fairness constraints, thereby ensuring that a target fairness level is achieved. We further introduce three different fairness criteria which can be efficiently tackled using FairRARI to compute fair PR vectors with the same asymptotic time-complexity as the original PR algorithm. Extensive experiments on real-world datasets showcase that FairRARI outperforms existing methods in terms of utility, while achieving the desired fairness levels across multiple vertex groups; thereby highlighting its effectiveness.

## 1. Introduction

Data-driven algorithms are being increasingly deployed in high-stakes decision-making systems, raising concerns about their potential social impact against certain demographic groups or individuals (Datta et al., 2015; Corbett-Davies et al., 2017; Chouldechova, 2017). Ensuring that algorithmic outcomes remain unbiased with respect to sensitive attributes has become a key challenge. The field of *algorithmic fairness* addresses this by designing fairness-aware mechanisms (Dwork et al., 2012; Feldman et al., 2015; Chierichetti et al., 2017; Kleinberg et al., 2018; Holstein et al., 2019), encompassing notions such as group-fairness, individual-fairness, and others (Mehrabi et al., 2021). While substantial progress has been made in supervised and unsupervised learning (Friedler et al., 2019; Chouldechova & Roth, 2020; Mehrabi et al., 2021), principled fairness-aware algorithms are still lacking for many fundamental graph machine learning (ML) primitives, despite recent efforts to address this gap (Dong et al., 2023; Saxena et al., 2024).

In particular, we focus on developing fairness-aware variants of the celebrated PageRank (PR) algorithm (Brin & Page, 1998; Langville & Meyer, 2006; Gleich, 2015). Originally designed for ranking web pages, PR has since been applied to a wide range of tasks, including link prediction (Liben-Nowell & Kleinberg, 2003), semi-supervised node classification (Berberidis et al., 2018), community detection (Whang et al., 2013), and graph representation learning (Gasteiger et al., 2019; Bojchevski et al., 2020). Our goal is to integrate different measures of group-fairness in PR, which seeks equitable treatment across demographic groups, and is a popular notion studied in a variety of graph ML tasks (Kleindessner et al., 2019; Rahmattalabi et al., 2019; Du et al., 2020; Dong et al., 2022; Kariotakis et al., 2025; Chowdhary et al., 2024),

Despite its widespread use, only recently has fairness in PR received attention. Prior studies (Avin et al., 2015; Espín-Noboa et al., 2022; Stoica et al., 2024) have shown that certain structural properties of graphs result in disparate allocation of PR scores across vertex subgroups defined by protected attributes, even when subgroups are balanced in composition or the level of homophily is moderate. These findings have motivated the development of fairness-aware PR variants (Tsioutsiouliklis et al., 2021; 2022; Rahman et al., 2019; Khajehnejad et al., 2022; Wang et al., 2026). Notwithstanding such efforts, important limitations remain: **(i)** At present, there is no principled optimization framework for computing fair PR vectors which are guaranteed to satisfy a target group-fairness requirement across multiple (i.e., $> 2$) vertex subgroups.
**(ii)** Prior work on fair PR has largely focused on a single notion of group-fairness introduced in (Tsioutsiouliklis et al.,

---

[1]Department of Electrical Engineering (ESAT-STADIUS), KU Leuven, Kasteelpark Arenberg 10, 3001 Leuven, Belgium. Correspondence to: Emmanouil Kariotakis <emmanouil.kariotakis@kuleuven.be>.

*Proceedings of the 43ʳᵈ International Conference on Machine Learning*, Seoul, South Korea. PMLR 306, 2026. Copyright 2026 by the author(s).

2021), whose adequacy has not been systematically examined, leaving alternative and potentially more appropriate group-fairness criteria largely underexplored.

*Our work addresses these gaps by putting forth a unified optimization framework for enforcing multiple notions of group-fairness in PageRank, with optimality guarantees.*

**Contributions:** We introduce FairRARI, a plug and play framework for computing group-fair PR vectors for various fairness criteria with provable guarantees. Our key insight is that in the absence of fairness, the PR vector admits a variational interpretation as the unique minimizer of an unconstrained strongly convex optimization problem. This enables group-fairness criteria to be incorporated into PR in a principled manner by enforcing appropriate constraints on the PR vector. We demonstrate that the resulting constrained optimization problem can be efficiently solved via a simple yet flexible fixed-point algorithm, where each iteration consists of a classical PR update followed by projection onto the fairness constraint set. As a result, we obtain fair PR solutions that satisfy target group-fairness requirements while remaining optimal with respect to the underlying PR objective. Our main contributions are:

- *Variational formulation of PageRank.* We reformulate original PR as the minimizer of a strongly convex quadratic function and show its equivalence to label spreading on graphs (Zhou et al., 2003). This perspective enables the principled incorporation of fairness constraints, yielding a new strongly convex constrained optimization problem.

- *A unified in-processing framework with guarantees.* We propose FairRARI, an in-processing algorithm that provably converges to the unique solution of the fair optimization problem for any closed and convex fairness constraint set, via simple fixed-point iterations, which decouple into two distinct updates. FairRARI subsumes the $\phi$-fairness criterion of (Tsioutsiouliklis et al., 2021) as a special case and naturally extends to the multi-group setting.

- *New and unified group-fairness criteria.* We study three group-fairness notions for PR. Beyond $\phi$-sum-fairness, which can lead to degenerate solutions with vertices attaining zero PR scores, we introduce a minimum-score fairness criterion that promotes more equitable score allocation within groups. We further show how these criteria can be combined into a unified formulation, offering fine-grained control over PR score allocation.

- *Linear-time projections.* Although FairRARI requires computing a projection at each iteration, we prove that for each proposed fairness criterion, the projection admits a linear-time solution. As a result, FairRARI enjoys the same asymptotic complexity as that of the

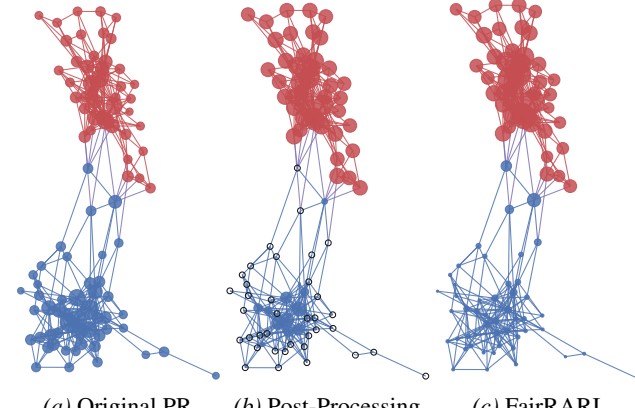

*(a)* Original PR    *(b)* Post-Processing    *(c)* FairRARI

*Figure 1.* Standard PageRank (PR) and fair PR vectors under $\phi$-fairness (Tsioutsiouliklis et al., 2021) with two vertex groups (red/blue) on the POLBOOKS dataset. Vertex size is proportional to the PR score produced by each method. Black circles indicate vertices with zero PR score. The standard PR vector (a) assigns $\phi_o^{\text{red}} = 0.47$ of the total PR mass to the red group. The target for the fair PR vectors is $\phi^{\text{red}} = 0.9$ of total mass for the red group. While both fair PR vectors in (b) and (c) attain this target, $37/49 \approx 75\%$ of blue vertices in (b) are set to zero, while $0\%$ of the blue vertices suffer such an undesirable outcome for FairRARI (c).

original PR algorithm (Gleich, 2015).

- *Comparison with post-processing baseline.* Similar to (Tsioutsiouliklis et al., 2021), we analyze a post-processing approach that minimally perturbs the original PR vector to satisfy a target fairness constraint, which we use as a bound on achievable utility. Although the solutions of FairRARI and post-processing are not directly comparable, through an analysis of the latter we show that it can produce qualitatively coarser results than FairRARI (see Fig. 1 where it produces outcomes where a large fraction of vertices receive zero PR scores).

- *Extensive empirical validation.* Experiments on real-world graphs demonstrate that FairRARI, by directly minimizing the true PR objective under fairness constraints, outperforms existing in-processing methods in terms of utility based on both PR scores and induced ranking. Moreover, it naturally supports multiple groups and various fairness criteria.

## 2. Related Work

The PageRank (PR) algorithm (Brin & Page, 1998) performs random walks on a graph to assign each vertex a score that reflects to its importance. However, it is not guaranteed to generate fair outcomes, as certain structural properties of graphs can result in unequal/biased allocation of PR scores across various vertex subgroups (Avin et al., 2015; Espín-Noboa et al., 2022; Stoica et al., 2024). As shown in (Avin

et al., 2015), imbalances in subgroup sizes among graph vertices, combined with varying levels of homophily (i.e., the tendency of vertices to connect with others from the same group), can lead to degree unfairness. As Tsioutsiouliklis et al. (2021) demonstrated, this in turn can cause PR to produce biased results.

Since PR is characterized by a teleportation vector, a probability transition matrix and the link structure, prior work in fairness-aware PR (Tsioutsiouliklis et al., 2021; 2022; Rahman et al., 2019; Khajehnejad et al., 2022; Wang et al., 2026) can be categorized according to the following design strategies: modifying (i) teleportation vector, (ii) transition matrix, and (iii) graph structure. However, these approaches come with drawbacks, which we discuss next.

Option (i) is considered in (Tsioutsiouliklis et al., 2021). The proposed FSPR (Fairness-Sensitive PR) algorithm computes a teleportation vector that provides $\phi$-fairness (i.e., the sum of PR scores of one group equals to a fraction $\phi \in (0, 1)$) by minimizing its distance from the original PR vector. However, it cannot handle the entire range $(0, 1)$ of $\phi$ values, but only a sub-interval determined by the graph structure. Option (ii) is the focus of Rahman et al. (2019); Khajehnejad et al. (2022). To mitigate potential group bias, the proposed FairWalk and CrossWalk modify the transition probabilities in a heuristic manner. Moreover, the approaches proposed by Wang et al. (2026) incorporate group-fairness into the PR algorithm by reweighting the transition probabilities in the underlying transition matrix. However, for all of those methods there is no guarantee of attaining a target fairness level. Tsioutsiouliklis et al. (2021) considered a combination of options (i) and (ii) - the LFPR (Locally-Fair PR) algorithms (LFPR$_\mathrm{N}$, LFPR$_\mathrm{U}$, and LFPR$_\mathrm{P}$). These ensure that $\phi$-fairness is attained by adjusting the transition matrix as well as the teleportation vector. The modifications involve altering the edge weights, and, in some cases, adding new edges. However, they lack a clear optimization formulation and offer no optimality guarantees. A fourth variant, LFPR$_\mathrm{O}$, modifies the transition matrix via a non-convex optimization problem, but without any optimality guarantees regarding the quality of the solution. Finally, Tsioutsiouliklis et al. (2022) studied option (iii) by quantifying the impact of adding and deleting edges on PR fairness, and proposed a link recommendation method to maximize fairness gains for a target vertex subgroup. However, this method does not guarantee that the desired level of fairness will be fulfilled.

Lastly, it is worth mentioning that our work differs significantly from prior work on fairness in ranking (Zehlike et al., 2017; Singh & Joachims, 2018; Geyik et al., 2019), since we optimize the PR scores of the vertices (and not the rankings explicitly) for attaining a target group-fairness level, which is based on equitable allocation of PR scores across groups.

# 3. Proposed Approach

## 3.1. Preliminaries

Consider a (un)directed graph $\mathcal{G} = (\mathcal{V}, \mathcal{E})$, with $n := |\mathcal{V}|$ vertices and $m := |\mathcal{E}|$ edges that is (strongly) connected. The PageRank (PR) algorithm models the action of a random walker visiting vertices in $\mathcal{G}$ in accordance with a $n \times n$ probability transition matrix $\mathbf{P}$, where each entry $P_{ij}$ denotes the probability of transitioning from vertex $j$ to vertex $i$. Let $\mathbf{A}$ and $\mathbf{D} \in \mathbb{R}^{n \times n}$ denote the adjacency and diagonal degree matrix of $\mathcal{G}$, respectively - for directed graphs, $\mathbf{D}$ denotes the out-degree matrix. The transition matrix $\mathbf{P}$ is formed by normalizing each column of $\mathbf{A}$ to sum to one, $\mathbf{P} := \mathbf{A}\mathbf{D}^{-1}$. Let $\mathbf{\Pi} := \mathrm{diag}(\boldsymbol{\pi})$, where $\boldsymbol{\pi}$ is the stationary distribution of $\mathbf{P}$. For an undirected $\mathcal{G}$ that is connected and non-bipartite, this reduces to $\mathbf{\Pi} = \frac{1}{2m}\mathbf{D}$. PR modifies the standard random walk on $\mathcal{G}$ by utilizing restarts, specified by a teleportation probability $\gamma$ and a teleportation vector $\mathbf{v}$. The output of the algorithm is the PR vector $\mathbf{p}_\mathrm{o}$. Since we use $\mathbf{x}$ as a network centrality measure, we set $\mathbf{v}$ to be the uniform vector. Then, each entry $x_i$ reflects the importance of vertex $i$ w.r.t. the global structure of $\mathcal{G}$.

Computing the original PR vector $\mathbf{p}_\mathrm{o}$ amounts to solving the following system of linear equations

$$\mathbf{x} = (1 - \gamma)\mathbf{P}\mathbf{x} + \gamma\mathbf{v}, \qquad (1)$$

which can be solved by applying the following fixed-point iterations (Gleich, 2015)

$$\mathbf{x}^{(t+1)} = (1 - \gamma)\mathbf{P}\mathbf{x}^{(t)} + \gamma\mathbf{v}, \qquad (2)$$

where $\mathbf{x}^{(0)} = \mathbf{v}$ or $\mathbf{x}^{(0)} = 0$. The sequence of generated iterates converges at a geometric rate to a unique fixed point; i.e., the PR vector $\mathbf{p}_\mathrm{o}$ (Gleich, 2015, Theorem 2.2). The method is also highly scalable, as the matrix-vector multiplication at each iteration can efficiently exploit the sparsity of the transition matrix $\mathbf{P}$. However, as noted earlier, the resulting PR vector may not satisfy a desired notion of group-fairness.

Interestingly, in the undirected case, PR iterations are intimately linked with label spreading (Zhou et al., 2003). If we perform a change of co-ordinates to obtain $\bar{\mathbf{x}}^{(t)} := \mathbf{\Pi}^{-\frac{1}{2}}\mathbf{x}^{(t)}$ and $\bar{\mathbf{v}} := \mathbf{\Pi}^{-\frac{1}{2}}\mathbf{v}$, then the PR iterations (2) become

$$\bar{\mathbf{x}}^{(t+1)} = (1 - \gamma)\tilde{\mathbf{A}}\bar{\mathbf{x}}^{(t)} + \gamma\bar{\mathbf{v}}, \qquad (3)$$

which is equivalent to performing label spreading in the transformed domain, with $\tilde{\mathbf{A}} := \mathbf{D}^{-\frac{1}{2}}\mathbf{A}\mathbf{D}^{-\frac{1}{2}}$ denoting the normalized adjacency matrix. This stems from the fact that $\tilde{\mathbf{A}}$ can be obtained by a similarity transformation applied to $\mathbf{P}$. Namely, if one defines $\mathbf{T} = \mathbf{\Pi}^{\frac{1}{2}}$, then $\mathbf{T}^{-1}\mathbf{P}\mathbf{T} = \mathbf{\Pi}^{-\frac{1}{2}}\mathbf{P}\mathbf{\Pi}^{\frac{1}{2}} = \mathbf{D}^{-\frac{1}{2}}\mathbf{A}\mathbf{D}^{-\frac{1}{2}} = \tilde{\mathbf{A}}$.

## 3.2. A Variational Interpretation of PageRank

Our starting point is the following variational characterization of the PR vector $\mathbf{p}_o$ (i.e., the solution of (1)).

**Proposition 3.1.** *The PR vector is the unique minimizer of the strongly convex quadratic optimization problem*

$$\min_{\mathbf{x}\in\mathbb{R}^n}\left\{f(\mathbf{x}):=\frac{1}{2}(1-\gamma)\mathbf{x}^\top\boldsymbol{\Pi}^{-1}\mathbf{L}_{\mathrm{rw}}\mathbf{x}+\frac{1}{2}\gamma\|\boldsymbol{\Pi}^{-\frac{1}{2}}(\mathbf{x}-\mathbf{v})\|^2\right\},\tag{4}$$

*where $\mathbf{L}_{\mathrm{rw}} := \mathbf{I} - \mathbf{P}$ is the random walk Laplacian of $\mathcal{G}$.*

The proof can be found in App. B.1, and holds for both undirected and directed graphs. For the latter case, we make an additional assumption that the Markov chain associated with $\mathbf{P}$ is time-reversible (Levin & Peres, 2017) [1]. The focus on the random walk Laplacian provides a unified framework for both undirected and directed graphs, as opposed to the variational formulation of PR proposed by Fountoulakis et al. (2019), which is limited to the undirected setting. For more details on the relationship between the two formulations, please refer to App. B.2.

The first component of the cost function $f(\cdot)$ is a smoothness-promoting term that encourages neighboring vertices to receive similar PR scores, while the second measures proximity to the teleportation vector $\mathbf{v}$. The teleportation parameter $\gamma$ controls the trade-off between these effects.

## 3.3. Problem Formulation

The variational interpretation of PR (4) allows the incorporation of fairness considerations in a principled manner. As mentioned previously, the PR vector; i.e., the solution of (4), is not guaranteed to satisfy a target fairness requirement. Let $\Delta_n$ denote the $n$-dimensional probability simplex, and $\mathcal{X} \subset \Delta_n$ denote any target fairness requirement which can be represented as a closed, convex set. Then, a natural means of enforcing these fairness constraints in PR is the following constrained convex optimization problem

$$\min\left\{f(\mathbf{x}) \text{ s.t. } \mathbf{x} \in \mathcal{X}\right\}.\tag{5}$$

This is the problem formulation that we consider in this paper. Since $f(\cdot)$ is strongly convex, it follows that the above problem has a unique minimizer (Beck, 2017, Theorem 5.25). Moreover, the requirement $\mathcal{X} \subset \Delta_n$ ensures that the minimizer is a *valid probability distribution* over the vertices of $\mathcal{G}$. Finally, by design, the solution is guaranteed to satisfy a target fairness criterion - by varying the choice of $\mathcal{X}$, different notions of group-fairness can be handled.

## 3.4. The FairRARI Algorithm

We now describe our approach to solving problem (5). An intuitive idea is to modify the classic PR fixed-point itera-

tions (2) by incorporating an extra step that projects iterates onto the set $\mathcal{X}$ in order to ensure feasibility. Formally, our algorithm is

$$\mathbf{y}^{(t+1)}=(1-\gamma)\mathbf{P}\mathbf{x}^{(t)}+\gamma\mathbf{v},\ \mathbf{x}^{(t+1)}=\mathrm{Proj}_\mathcal{X}(\mathbf{y}^{(t+1)}),\ (6)$$

where we make use of the *projection operator* onto $\mathcal{X}$, $\mathrm{Proj}_\mathcal{X}(\mathbf{y}) := \mathrm{argmin}_{\mathbf{x}\in\mathcal{X}}\|\mathbf{x} - \mathbf{y}\|_2^2$. Hence, fairness is guaranteed by modifying the PR vector directly, while keeping the transition matrix $\mathbf{P}$ and the teleportation vector $\mathbf{v}$ fixed. We denote (6) as the **FairRARI** iterations. At this point, it is natural to ask whether the iterations converge to a unique fixed point, and whether this actually corresponds to the solution of (5). Our following pair of results shows that this is indeed the case. Please see App. B.3 and B.4 for the proofs.

**Theorem 3.2.** *Fix a graph $\mathcal{G}$ (directed or undirected) and let $\mathcal{X} \subset \Delta_n$ denote any closed convex set. Then, the FairRARI iterations (6) converge at a geometric rate to a unique fixed point.*

**Theorem 3.3.** *Let the transition matrix $\mathbf{P}$ correspond to a time-reversible Markov chain. Then, the vector $\mathbf{x}^*$ is the minimizer of problem (5), if and only if it is the unique fixed-point of the FairRARI iterations (6).*

**Remark 1.** Theorem 3.2 asserts that the FairRARI iterations (6) always converge to a unique fixed point for any graph $\mathcal{G}$, while Theorem 3.3 provides a sufficient condition on $\mathcal{G}$ under which the fixed point of (6) equals the minimizer of problem (5). For undirected graphs, the transition matrix $\mathbf{P}$ is time-reversible by construction, and hence Theorem 3.3 always applies. In the directed setting, the assumption that $\mathbf{P}$ is time-reversible can be restrictive. However, it can be weakened to the condition that the combined Markov chain $\tilde{\mathbf{P}} := (1 - \gamma)\mathbf{P} + \gamma\mathbf{v}\mathbf{e}^\top$, with $\mathbf{v} = \frac{1}{n}\mathbf{e}$, is time-reversible. This is a more justifiable assumption since the incorporation of teleportation promotes time-reversibility of $\tilde{\mathbf{P}}$, even when $\mathbf{P}$ does not satisfy this property (for a more detailed discussion, please refer to App. B.5). Furthermore, in App. G.2, we show that for the real-world directed graphs used in our experiments, the matrix $\tilde{\mathbf{P}}$ is very close to being time-reversible, even for a small teleportation parameter $\gamma = 0.15$. Assuming that $\tilde{\mathbf{P}}$ is time-reversible in the directed setting, a result analogous to that of Theorem 3.3 can be established which shows that the fixed point of the FairRARI iterations (6) correspond to a minimizer of a constrained convex quadratic optimization problem, defined in terms of the random walk Laplacian $\tilde{\mathbf{L}}_{\mathrm{rw}} := \mathbf{I} - \tilde{\mathbf{P}}$ (for the proof, please refer to App. B.5).

**Remark 2.** For directed graphs, it may appear on first glance that the stationary distribution $\boldsymbol{\pi}$ needs to be pre-computed, since the cost function of (5) depends on $\boldsymbol{\pi}$. However, this is not the case. Note that the FairRARI iterations do not require foreknowledge of $\boldsymbol{\pi}$, and Theorems 3.2

---

[1] For undirected graphs, this condition is automatically satisfied.

and 3.3 assert that the fixed point of (6) yields the solution of (5).

**Remark 3.** The FairRARI iterations (6) can be conceptually viewed as the following form of projected gradient descent applied on the optimization problem (5)

$$\mathbf{y}^{(t+1)} = \mathbf{x}^{(t)} - \mathbf{\Pi}\nabla f(\mathbf{x}^{(t)}), \quad \mathbf{x}^{(t+1)} = \text{Proj}_{\mathcal{X}}(\mathbf{y}^{(t+1)}). \quad (7)$$

The PR step performs preconditioned gradient descent on the smooth, strongly convex objective $f(\cdot)$, and the projection step ensures feasibility w.r.t the fairness set $\mathcal{X}$. Hence, in sharp contrast with the prevailing methods of (Rahman et al., 2019; Khajehnejad et al., 2022; Wang et al., 2026), and the LFPR family of methods (Tsioutsiouliklis et al., 2021), not only does FairRARI guarantee that a target fairness criterion is fulfilled, but the solution (i.e., the fixed point) is also optimal in a certain sense.

***Complexity Analysis:*** Given a desired relative tolerance value $\epsilon$, from Theorem 3.2 it follows that the number of iterations $t$ required to ensure that $\|\mathbf{x}^{(t)} - \mathbf{x}^*\|^2 / \|\mathbf{x}^{(0)} - \mathbf{x}^*\|^2 \leq \epsilon$, is $O(\log(1/\epsilon)/\log(1/(1-\gamma)))$. The per-iteration complexity of the iterations (6) can be determined by analyzing each step - the first step requires computing the matrix-vector multiplication $\mathbf{P}\mathbf{x}^{(t)}$, which incurs $O(m)$ complexity, followed by adding $\gamma\mathbf{v}$ to the result. The overall complexity is then $O(m + n)$. The second step involves computing the Euclidean projection onto the fairness constraint $\mathcal{X}$, which requires solving a convex optimization problem of the form

$$\min_{\mathbf{x} \in \mathcal{X}} \|\mathbf{x} - \mathbf{y}\|_2^2. \quad (8)$$

The main computational bottleneck lies in solving this problem efficiently. Although it is a convex QP solvable by off-the-shelf solvers, these ignore the specific structure of the constraint set $\mathcal{X}$, leading to worst-case complexity of $O(n^3)$ (Goldfarb & Liu, 1990). In the following sections, we introduce specific fairness criteria and show that, by exploiting the structure of $\mathcal{X}$, each projection can be solved far more efficiently, in *linear-time*.

# 4. Fairness Criteria

## 4.1. Proposed Criteria

Consider a partition of the vertices of $\mathcal{G}$ into $K$ vertex groups $\{\mathcal{S}_k\}_{k=1}^K$, each of which represents a different type of a sensitive attribute (e.g., political leaning, race, or other demographic information). Let $n_k := |\mathcal{S}_k|$ denote the number of vertices in group $\mathcal{S}_k$. We consider various convex group-fairness constraints $\mathcal{X} \subset \Delta_n$ which regulate how the PR scores are assigned across different vertex groups $\{\mathcal{S}_k\}_{k=1}^K$.

○ ***$\phi$-sum-fairness:*** Given a set of target fairness levels $\{\phi_k\}_{k=1}^K$ satisfying $\phi_k \geq 0, \sum_{k=1}^K \phi_k = 1$, the sum of

the PR scores for each group $k$ equals $\phi_k$. These constraints can be compactly denoted by the set

$$\Delta_{\boldsymbol{\phi}}(\mathcal{V}) := \{\mathbf{x} \geq \mathbf{0} : \mathbf{e}_{\mathcal{S}_k}^\top \mathbf{x} = \phi_k, \forall k \in [K]\}, \quad (9)$$

where $[K] := \{1, \ldots, K\}$ and $\mathbf{e}_{\mathcal{S}_k} \in \{0, 1\}^n$ being the binary indicator vector of the group $\mathcal{S}_k$, i.e., $\mathbf{e}_{\mathcal{S}_k}(i) = 1$ if and only if vertex $i \in \mathcal{S}_k$. Each linear constraint ensures that the PR mass $\sum_{i \in \mathcal{S}_k} x_i$ of a vertex group $\mathcal{S}_k$ equals a fraction $\phi_k$ of the total sum of the PR scores of $\mathbf{x}$ (across all groups). A vector $\mathbf{x}$ which satisfies the constraints (9) is said to be $\phi$-sum-fair. By varying the parameters $\{\phi_k\}_{k=1}^K$, various parity based fairness policies can be implemented to prevent the PR mass being allocated unevenly (i.e., unfairly) across different groups. For example, choosing $\phi_k = \frac{n_k}{n}, \forall k \in [K]$ implies that the share of PR scores allocated to group $\mathcal{S}_k$ is proportional to the share of vertices $n_k$ in the population, a property known as *demographic parity* (Dwork et al., 2012). The case of $K = 2$ was first studied in (Tsioutsiouliklis et al., 2021), where it was referred to as $\phi$-fairness. Here, we adopt a slightly different designation in order to better distinguish $\Delta_{\boldsymbol{\phi}}(\mathcal{V})$ from the subsequent fairness criterion. To the best of our knowledge, at present, there is no algorithm which can guarantee $\phi$-sum-fairness for $K > 2$ groups.

○ ***$\alpha$-min-fairness:*** The group-fairness constraints (9) ensure that the total PR mass of a group $\mathcal{S}_k$ equals the target level $\phi_k$, but they allow this mass to be distributed arbitrarily within the group. As a result, some vertices in $\mathcal{S}_k$ may receive very low, or even zero, PR scores (see Fig. 1), an outcome undesirable in practice, as it removes vertices from consideration in downstream tasks. This observation motivates the introduction of fairness constraints that regulate not only aggregate group mass, but also the minimum score assigned to individual vertices. Hence, we can consider an alternative notion of group-fairness motivated by the *Rawlsian principle of social justice* (Rawls, 2020). Given a fixed subset of vertices $\mathcal{A}_k \subseteq \mathcal{S}_k$ of group $k \in [K]$, the goal is to ensure that *every* vertex in $\mathcal{A}_k$ is allocated a minimum PR score $\alpha_k \geq 0$. These constraints can be expressed by the set

$$\Delta^{\boldsymbol{\alpha}}(\mathcal{A}) := \{\mathbf{x} \geq \mathbf{0} : \mathbf{e}^\top \mathbf{x} = 1, \min_{i \in \mathcal{A}_k} x_i \geq \alpha_k, \forall k \in [K]\}, \quad (10)$$

where $\mathbf{e}$ is the $n$-dimensional all-ones vector, and the target fairness parameters $\{\alpha_k\}_{k=1}^K$ dictate the minimum PR score assigned to each group. If $\mathcal{A}_k \equiv \mathcal{S}_k$ then we ensure that every vertex in each group $\mathcal{S}_k$ is allocated a minimum PR score $\alpha_k$. Let $\boldsymbol{\alpha} := (\alpha_1, \ldots, \alpha_K)^\top$ denote the vector of target fairness levels. A PR vector satisfying the above constraints (which are linear in $\mathbf{x}$) is said to be $\boldsymbol{\alpha}$-min-fair. In order to guarantee that the set $\Delta^{\boldsymbol{\alpha}}(\mathcal{A})$ is feasible, it suffices to select $\{\alpha_k\}_{k=1}^K$ such that $\sum_{k=1}^K \alpha_k |\mathcal{A}_k| \leq 1$.

○ ***$\phi$-sum + $\alpha$-min fairness:*** The most general form of group-fairness considered combines the two aforementioned notions into a single criterion. Here, each group $\mathcal{S}_k$ has PR

scores that sum to a target value $\phi_k$, while vertices belonging to a target subgroup $\mathcal{A}_k \subseteq \mathcal{S}_k$ have a PR score at least $\alpha_k$. These constraints can be expressed by the set

$$\Delta_\phi^\alpha(\mathcal{V},\mathcal{A}):=\{\mathbf{x}\geq\mathbf{0}:\mathbf{e}_{\mathcal{S}_k}^\top\mathbf{x}=\phi_k,\min_{i\in\mathcal{A}_k}x_i\geq\alpha_k,\forall k\in[K]\}. \quad(11)$$

The above constraints offer finer-grained control of the allocation of PR scores across the groups $\{\mathcal{S}_k\}_{k=1}^K$ compared to (9) or (10) fairness alone. For example, by setting $\boldsymbol{\alpha}=\mathbf{0}$, (11) reduces to (9). Similarly, on replacing the $K$ sum-to-$\phi_k$ constraints on $\mathbf{x}$ by the looser sum-to-1 constraint, we obtain (10). The parameters $\boldsymbol{\phi}$ and $\boldsymbol{\alpha}$ are chosen to satisfy $\sum_{k=1}^K \phi_k = 1$ and $\alpha_k|\mathcal{A}_k| \leq \phi_k, \forall\, k \in [K]$.

## 4.2. Efficient Projections

For the proposed fairness criteria, we show that by leveraging the structure of each constraint set, we can significantly reduce the complexity of the projection step. Hence, when we refer to $\mathcal{X}$, we will refer to one of the sets that we defined for each fairness criterion. For $\phi$-sum-fairness, we have $\Delta_\phi(\mathcal{V})$, for $\boldsymbol{\alpha}$-min-fairness, $\Delta^\alpha(\mathcal{A})$, and for $\phi$-sum + $\boldsymbol{\alpha}$-min-fairness, $\Delta_\phi^\alpha(\mathcal{V},\mathcal{A})$. Given a vector $\mathbf{y}$ that we wish to project onto any of the aforementioned $\mathcal{X}$, the Euclidean projection problem corresponds to finding the (unique) minimizer of problem (8). Our subsequent results characterize the optimal solution of the problem, for each constraint set $\mathcal{X}$, via its KKT conditions. Please refer to App. C.1 for proofs and details.

**Theorem 4.1.** *Let* $\mathbf{x}^*$ *denote an optimal solution of problem* (8), *with* $\mathcal{X} = \Delta_\phi(\mathcal{V})$. *Then, it is given by*

$$\mathbf{x}_{\mathcal{S}_k}^* = \max\{0, \mathbf{y}_{\mathcal{S}_k} - \lambda_k^*\mathbf{e}_{\mathcal{S}_k}\}, \forall\, k \in [K], \quad(12)$$

*where the variables* $\lambda_k^*, \forall\, k \in [K]$, *satisfy*

$$\sum_{i\in\mathcal{S}_k} \max\{0, y_i - \lambda_k^*\} = \phi_k. \quad(13)$$

In (12), the vectors $\mathbf{x}_{\mathcal{S}_k}^*$ and $\mathbf{y}_{\mathcal{S}_k}$ denote the $n_k \times 1$ subvectors of $\mathbf{x}^*$ and $\mathbf{y}$ indexed by $\mathcal{S}_k$ respectively, and the max-operator is applied element-wise. Meanwhile, the variables $\{\lambda_k^*\}_{k=1}^K$ can be viewed as the optimal dual variables corresponding to the linear constraints of $\Delta_\phi(\mathcal{V})$. It is evident that once the dual variables are computed from (13), they completely determine the optimal solution $\mathbf{x}^*$ via (12). Further examination of (13) also reveals that each $\lambda_k^*$ can be computed independently of the others to determine $\mathbf{x}_{\mathcal{S}_k}^*$. Hence, the computation of $\mathbf{x}^*$ can be trivially parallelized. Each dual variable $\lambda_k^*$ can be computed using binary search. Please refer to App C.1 for details.

***Complexity:*** For sequential execution, with fixed tolerance values $\{\epsilon_k\}_{k=1}^K$, one for each $\lambda_k^*$, the total complexity is

**Algorithm 1** The FAIRRARI framework
___
> **Input:** $\mathcal{X}, \mathcal{G} = (\mathcal{V}, \mathcal{E}), \gamma, \mathbf{v}, \epsilon$
> **Output:** Fair PageRank vector
> **Initialize:** transition matrix $\mathbf{P}$, teleportation vector $\mathbf{v}$
> **while** $\|\mathbf{x}^{(t)} - \mathbf{x}^{(t-1)}\| \geq \epsilon$ **do**
> $\quad\mathbf{y}^{(t+1)} \leftarrow (1 - \gamma)\mathbf{P}\mathbf{x}^{(t)} + \gamma\mathbf{v}$
> $\quad\mathbf{x}^{(t+1)} \leftarrow$ FAIR-PROJECTION$(\mathbf{y}^{(t+1)}, \mathcal{X}, \mathcal{G})$
> **end while**
> **return** $\mathbf{x}^* \leftarrow \mathbf{x}^{(t+1)}$
___

$\sum_{k=1}^K O(n_k) = O(n)$. In contrast, for parallel execution, the complexity is determined by the largest group, i.e., $O(n_k)$, where $n_k = \max_{i\in[K]} n_i$.

**Theorem 4.2.** *Let* $\mathbf{x}^*$ *denote an optimal solution of problem* (8), *with* $\mathcal{X} = \Delta^\alpha(\mathcal{A})$. *Then, it is given by*

$$x_i^* = \begin{cases} \max\{\alpha_k, y_i - \lambda^*\}, & i \in \mathcal{A}_k \\ \max\{0, y_i - \lambda^*\}, & i \in \bar{\mathcal{A}}_k \end{cases}, \forall\, k \in [K], \quad(14)$$

*where the variable* $\lambda$ *satisfies*

$$\sum_{k=1}^K \left\{ \sum_{i\in\mathcal{A}_k}\max\{\alpha_k, y_i-\lambda^*\} + \sum_{i\in\bar{\mathcal{A}}_k}\max\{0, y_i-\lambda^*\} \right\} = 1. \quad(15)$$

***Complexity:*** For a fixed tolerance $\epsilon$, the complexity of the Fair Projections is linear to the number of vertices of the graph, i.e., $O(n)$, since for the $\boldsymbol{\alpha}$-min-fairness case we need to perform only one bisection per projection.

**Theorem 4.3.** *Let* $\mathbf{x}^*$ *denote an optimal solution of problem* (8), *with* $\mathcal{X} = \Delta_\phi^\alpha(\mathcal{V},\mathcal{A})$. *Then, it is given by*

$$x_i^* = \begin{cases} \max\{\alpha_k, y_i - \lambda_k^*\}, & i \in \mathcal{A}_k \\ \max\{0, y_i - \lambda_k^*\}, & i \in \bar{\mathcal{A}}_k \end{cases}, \forall\, k \in [K], \quad(16)$$

*where the variables* $\lambda_k^*, \forall\, k \in [K]$, *satisfy*

$$\sum_{i\in\mathcal{A}_k}\max\{\alpha_k, y_i-\lambda_k^*\} + \sum_{i\in\bar{\mathcal{A}}_k} \max\{0, y_i-\lambda_k^*\} = \phi_k. \quad(17)$$

***Complexity:*** The complexity of Fair Projections in this case is identical to that of $\phi$-sum-fairness; $O(n)$ for sequential execution, and $O(n_k)$, with $n_k = \max_{i\in[K]} n_i$, for parallel.

## 4.3. The FairRARI Framework

Having described the fairness criteria, we present the pseudocode of the FairRARI framework in Algorithm 1. Based on the choice of the constraint set $\mathcal{X}$, the FAIR-PROJECTION subroutine changes and handles either $\phi$-sum-fairness (9), $\boldsymbol{\alpha}$-min-fairneess (10) or their combination (11) (for pseudocode of the FAIR-PROJECTION subroutine, please refer to App. D). We now finalize the time complexity of the framework. As we have shown in Section 3, the complexity of the

first step of FairRARI iterations (6) is $O(m + n)$. Additionally, for the second part, we have shown that the worst-case complexity of the projections is $O(n)$, resulting in an overall per-iteration complexity of $O(m + n)$. Hence, the final complexity figure after $T$ iterations is $O((m+n) \cdot T)$, where $T = O\left(\frac{\log(1/\epsilon)}{\log(1/(1-\gamma))}\right)$. Hence, *FairRARI exhibits the same asymptotic time complexity as the standard (non-fair) PR algorithm.*

## 5. Analysis of Post-Processing

We consider a post-processing approach that modifies $\mathbf{p}_o$ (the original PR vector) to satisfy a target fairness constraint $\mathcal{X}$. The goal is to find the smallest $\ell_2$-perturbation of $\mathbf{p}_o$ that makes it feasible for $\mathcal{X}$. This can be framed as projecting $\mathbf{p}_o$ onto $\mathcal{X}$, which entails solving the following problem

$$\min_{\mathbf{x} \in \mathcal{X}} \|\mathbf{x} - \mathbf{p}_o\|_2^2. \tag{18}$$

When $\mathcal{X}$ is closed and convex, a minimizer of the problem (18) always exists, and is unique for any given vector $\mathbf{p}_o$ (Beck, 2017, Theorem 6.25). The solution provides a lower bound on the magnitude of the modification required by any in-processing approach to satisfy a target fairness level, since it is agnostic to the structure of $\mathcal{G}$. Hence, it may not be attainable by an in-processing algorithm. Such an approach was originally proposed in Tsioutsiouliklis et al. (2021), who presented an algorithm for problem (18) for the case where $\mathcal{X}$ corresponds to $\phi$-sum fairness with two groups. However, their proposed algorithm is not guaranteed to solve (18). Here, we provide a rigorous analysis of the solution of (18) for general $\mathcal{X}$ via the KKT conditions, and contrast it with the in-processing solution produced by FairRARI. The main difference is that FairRARI jointly addresses utility and fairness via (5), while the other one-shot projects the *final* PR vector $\mathbf{p}_o$, onto the fairness set $\mathcal{X}$ without regard to the graph structure. Our analysis reveals that this difference causes post-processing to induce a markedly coarser modification of $\mathbf{p}_o$, whereas FairRARI preserves smoother score distributions. This is formalized by the following result (see App. B.6 for the proof).

**Theorem 5.1.** *Let $\mathbf{p}_o$ denote the original PageRank vector on $\mathcal{G}$, and let $\mathcal{X}$ the closed convex set defined by inequality and equality constraints $\mathcal{X} = \{\mathbf{x} \in \mathbb{R}^n \mid w_i(\mathbf{x}) \leq 0, \ i = 1, \ldots, p; \ u_j(\mathbf{x}) = 0, \ j = 1, \ldots, q\}$. Then, the optimal solution of the post-processing problem (18) is characterized by the KKT conditions as*

$$\mathbf{x}^* = \mathbf{p}_o - \frac{1}{2} z(\mathbf{x}^*), \tag{19}$$

*whereas the solution of the FairRARI problem (5) satisfies*

$$\mathbf{x}^* = \mathbf{p}_o - [\mathbf{I} - (1-\gamma)\mathbf{P}]^{-1} \mathbf{\Pi} z(\mathbf{x}^*), \tag{20}$$

*where $z(\mathbf{x}) = \sum_{i=1}^p \lambda_i^* \nabla w_i(\mathbf{x}) + \sum_{j=1}^q \mu_j^* \nabla u_j(\mathbf{x})$, with $\lambda_i^*$ and $\mu_j^*$ denoting the optimal dual variables associated with the inequality and equality constraints, respectively.*

**Remark 4.** Eq. (19) shows that post-processing modifies $\mathbf{p}_o$ through a direct, graph-agnostic correction determined solely by the active fairness constraints. In contrast, Eq. (20) reveals the inherently in-processing nature of FairRARI. Re-expressing the matrix $[\mathbf{I} - (1-\gamma)\mathbf{P}]^{-1} = \sum_{j=0}^{\infty}(1-\gamma)^j \mathbf{P}^j$, shows that FairRARI corrects $\mathbf{p}_o$ via a graph diffusion term to arrive at the solution of (5). Further insight is obtained by considering the special case of $\phi$-fairness with two groups (Tsioutsiouliklis et al., 2021), for which we derive a semi-analytical form of (19). The proof is provided in App. B.7.

**Corollary 5.2.** *Consider $\phi$-fairness for two groups, $\mathcal{S}_1$ and $\mathcal{S}_2$, with the constraint that the solution $\mathbf{x}^*$ satisfies $\sum_{i \in \mathcal{S}_1} \mathbf{x}^*(i) = \phi$. Define $\phi_o := \sum_{i \in \mathcal{S}_1} \mathbf{p}_o(i)$ and w.l.o.g. assume that $\phi - \phi_o = \epsilon > 0$. Then, (19) can be expressed as*

$$\mathbf{x}^*(i) = \begin{cases} \mathbf{p}_o(i) + \frac{\epsilon}{|\mathcal{S}_1|}, & i \in \mathcal{S}_1 \\ \mathbf{p}_o(i) - \frac{\epsilon - c}{|\mathcal{S}_2^+|}, & i \in \mathcal{S}_2^+ := \left\{i \in \mathcal{S}_2 : \mathbf{p}_o(i) \geq \frac{\epsilon - c}{|\mathcal{S}_2^+|}\right\} \\ 0, & i \in \mathcal{S}_2^- := \left\{i \in \mathcal{S}_2 : \mathbf{p}_o(i) < \frac{\epsilon - c}{|\mathcal{S}_2^+|}\right\} \end{cases} \tag{21}$$

*where $c = \sum_{i \in \mathcal{S}_2^-} \mathbf{p}_o(i)$.*

This corollary illustrates that the post-processing method effects a coarse modification of $\mathbf{p}_o$, as it modifies PR scores uniformly in both groups, without accounting for the graph's structure (as opposed to FairRARI). As a result, it may enforce fairness at the cost of assigning zero PR scores to certain vertices. This behavior is evident in Fig. 1, where the post-processing method zeros out 75% of the blue vertices, whereas FairRARI assigns a positive score to each vertex. In the experiments (Section 6), utility is evaluated in terms of the distance from the original PR vector $\mathbf{p}_o$. Under this utility measure, post-processing achieves the smallest possible deviation from $\mathbf{p}_o$, since it directly adjusts $\mathbf{p}_o$ without considering the graph structure. For this reason, we use it only as a benchmark, providing a bound on the distance to $\mathbf{p}_o$ that an in-processing alternative can achieve.

## 6. Experiments

### 6.1. Experimental Setup

*Datasets - Code:* We evaluate the performance of our methods on 22 real-world datasets (see App. E for more information). Implementation details are provided in App. G.1, and the code is publicly available on Github[2].

*Baselines:* We use the FSPR and LFPR algorithms of Tsioutsiouliklis et al. (2021) as baselines. Fair-Walk (Rahman et al., 2019), CrossWalk (Khajehnejad

[2] https://github.com/ekariotakis/FairRARI

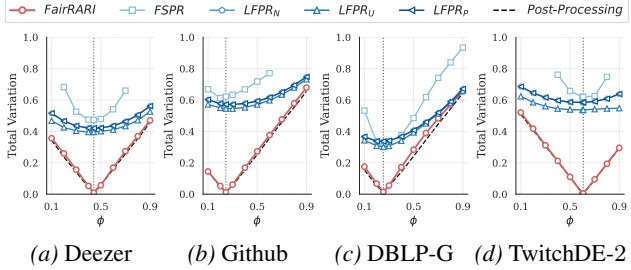

*(a)* Deezer    *(b)* Github    *(c)* DBLP-G    *(d)* TwitchDE-2

*Figure 2.* Comparing Fair PR solutions of prior methods and FairRARI (Ours), on different datasets for $\phi$-sum-fairness with 2 groups. The figures showcase the TV utility of each solution for different fairness levels $\phi$ (lower the better). Vertical dotted lines: $\phi_o$.

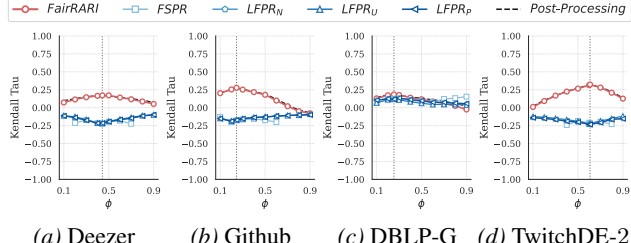

*(a)* Deezer    *(b)* Github    *(c)* DBLP-G    *(d)* TwitchDE-2

*Figure 3.* Comparing Fair PR solutions of prior methods and FairRARI (Ours), on different datasets for $\phi$-sum-fairness with 2 groups. The figures showcase the Kendall Tau coefficient of each solution for different fairness levels $\phi$ (higher the better). Vertical dotted lines: $\phi_o$.

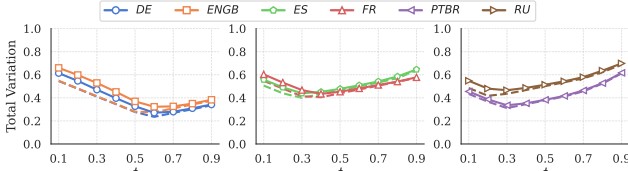

*Figure 4.* Showcasing the TV of the FairRARI solutions for different fairness levels $\phi$, on Twitch datasets with 4 groups. The dashed lines represent the post-processing solution for each dataset.

et al., 2022), FairGD and AdaptGD proposed by Wang et al. (2026) are excluded, as they substantially violate the target fairness level (see App. G.4.5). For detailed description of these algorithms, see App. F. Finally, we use the post-processing approach as a bound to benchmark performance of in-processing methods.

***Evaluation Criteria:*** We consider two evaluation criteria. **(i):** Let $\mathbf{p}_o$ denote the original PR vector and $\mathbf{p}_F$ denote a candidate fair PR vector. Since both are probability distributions, we use the total variation (TV) distance, $\mathrm{TV}(\mathbf{p}_o, \mathbf{p}_F) := \frac{1}{2}\|\mathbf{p}_o - \mathbf{p}_F\|_1$, to measure utility. Results based on the $\ell_2$-distance utility of (Tsioutsiouliklis et al., 2021) are reported in App. G.4.3. **(ii):** To evaluate the rankings induced by the fair PR vectors $\mathbf{p}_F$, we use the Kendall Tau rank correlation coefficient (Kendall, 1938), computed between the rankings induced by the pair $(\mathbf{p}_o, \mathbf{p}_F)$, with higher values indicating stronger agreement between the rankings induced by $\mathbf{p}_o$ and $\mathbf{p}_F$. For a comparison based on the top-$k$ rankings instead of the entire ranking list, please refer to App. G.4.4.

### 6.2. Experimental Evaluation

***Timing Results:*** Briefly, our experiments confirm that FairRARI asymptotically shares the same time complexity as PR, as they show that FairRARI incurs at most $10\%$ increase in time complexity compared to the original PR algorithm in the worst case, in contrast to prior methods that can exceed $5000\%$. For the timing results, refer to App. G.3.

***$\phi$-sum-fairness:*** For datasets with 2 groups, we set the target fairness vector to be $\phi = (\phi, 1 - \phi)$. For multiple groups, we set $\phi_1 = \phi$ for the first group, and we split the balance $1 - \phi$ equally among the rest. Fig. 2 illustrates a comparative analysis with prior methods across multiple real-world datasets. For various values of $\phi$, FairRARI consistently yields a significantly lower TV, very close to the lower bound, indicating that the resulting fair PR distribution is much closer to that of the original PR. Notably, when $\phi$ is equal to $\phi_o$ (vertical dotted lines), FairRARI attains

zero TV, in contrast to the prior methods, which incur a much higher cost, typically around $0.5$. This underscores the effectiveness of our approach, since when the original PR solution already satisfies the desired fairness level, it is naturally the optimal solution.

Fig. 3 complements the above analysis by measuring the agreement between the rankings induced by the fair PR vectors $\mathbf{p}_F$ and the original PR vector $\mathbf{p}_o$, using the Kendall Tau rank correlation coefficient. Higher values indicate closer agreement. Across most fairness levels and datasets, FairRARI achieves the highest Kendall Tau scores, significantly outperforming prior methods. Overall, these results highlight the advantage of directly optimizing the true PR objective under fairness constraints, enabling FairRARI to achieve low deviation from the original PR vector while closely preserving the original ranking.

In the multi-group setting, FairWalk (Rahman et al., 2019), CrossWalk (Khajehnejad et al., 2022) and the methods of Wang et al. (2026) cannot attain the target fairness level, as explained before. In contrast, FairRARI effectively addresses this setting by satisfying the fairness constraints while achieving TV values that are close to the post-processing solution. Notably, FairRARI achieves lower TV on the Twitch datasets even in the more challenging setting with 4 groups, compared to prior methods on the easier 2-group setting. As shown in Fig. 4, the TV for the TwitchDE dataset with 4 groups is *consistently lower* than that of the prior methods on TwitchDE-2 with 2 groups (Fig. 2).

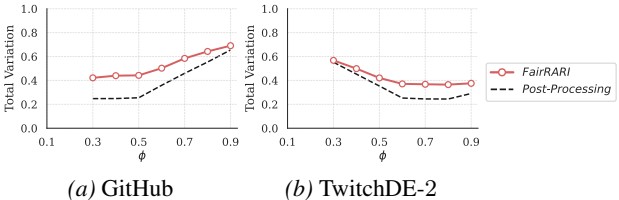

*Figure 5.* Showcasing the TV of FairRARI and Post-Processing solutions on the $\phi$-sum + $\alpha$-min-fair problem.

$\phi$-***sum*** + $\alpha$-***min-fairness:*** As illustrated in Fig. 1, enforcing $\phi$-sum-fairness can result in some vertices receiving zero scores. This observation motivates the need of this more stringent criterion, which additionally enforces a minimum score threshold for certain vertices. Specifically, we consider 2 groups, where we select $\mathcal{A}$ to be a subset of $\mathcal{S}_1$, containing the top $|\mathcal{S}_1|/100$ vertices of $\mathcal{S}_1$, in terms of vertex degree. In these experiments, we fix the value of $\alpha$ and vary $\phi$. More precisely, we set $\alpha = 0.25/|\mathcal{A}|$, and since it should always hold that $\alpha|\mathcal{A}| < \phi \Rightarrow \phi > 0.25$, we vary $\phi$ from 0.3 to 0.9.

In Fig. 5, we present the TV achieved by FairRARI and compare it against the lower bound obtained via the post-processing method. Since this is a more stringent fairness constraint, the bound remains far from zero, but FairRARI achieves TV values that are close to it. Notably, there are cases where the TV achieved by FairRARI, under the combined $\phi$-sum + $\alpha$-min-fairness constraints, is lower than that of prior methods that enforce the simpler $\phi$-sum-fairness constraint, such as on the TwitchDE-2 dataset.

For experiments on additional datasets, refer to App. G, from which the same conclusions can be made as in this section.

## 7. Conclusion

We introduced FairRARI, a principled and flexible plug and play framework for computing fair PageRank (PR) vectors under various group-fairness constraints. FairRARI guarantees convergence to fair solutions via fixed-point iterations based on projections onto convex sets. From a variational perspective, we showed that FairRARI is a form of projected gradient descent for computing the minimizer of a strongly convex optimization problem with fairness constraints. We instantiated FairRARI with three group-fairness criteria, a generalization of $\phi$-fairness to multiple groups, and two new criteria enabling finer-grained score control within groups. All these constraints can be efficiently tackled by FairRARI via linear-time projections, thereby retaining the original PR algorithm's asymptotic computational complexity. Experiments on real-world datasets demonstrate FairRARI's strong empirical performance, outperforming existing methods in terms of PR utility and ranking, while ensuring fairness across diverse scenarios.

## Acknowledgments

Supported by the KU Leuven Special Research Fund BOF/STG-22-040.

## Impact Statement

The PageRank (PR) algorithm allocates centrality scores to vertices in a graph via random walks and can perpetuate or amplify structural biases that exist across vertex subgroups in graphs. This work advances the field of algorithmic fairness on graphs by developing methods for computation of PR vectors under group-fairness constraints defined by sensitive vertex attributes. Via such an approach, we aim to promote fairness in centrality scores while preserving the utility of standard PR.

While this work focuses on group-fairness criteria and captures three distinct group-fairness notions, we acknowledge that it does not explicitly address other relevant notions of fairness, such as individual and popularity fairness. Accordingly, our results should be interpreted within this scope: the objective of this work is to improve the trade-off between group-fairness constraints and PR utility, and our findings should not be directly extrapolated to other notions of fairness. We view our work as a first step and anticipate future research that extends our framework to incorporate additional notions of fairness.

We hope this contribution supports the development of more equitable graph-learning systems and motivates further research in algorithmic fairness for graph machine learning.

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

# A. Supporting Results

In this section, we compile a collection of known results which will prove useful in the proofs of the theorems in the subsequent sections.

## A.1. Basics of fixed-point operators

This overview is primarily based on the content of (Ryu & Yin, 2022, Chapter 2). Let $T : \mathbb{R}^n \to \mathbb{R}^n$ denote a single valued operator (i.e., a function) with domain $\text{dom}(T) = \{\mathbf{x} : T(\mathbf{x}) \neq \emptyset\}$. We say that a point $\mathbf{x}$ is a fixed point of $T(\cdot)$ if it satisfies $\mathbf{x} = T(\mathbf{x})$. The set of fixed points of $T(\cdot)$ are denoted as $\text{Fix}(T) = \{\mathbf{x} : \mathbf{x} = T(\mathbf{x})\}$. For an $L > 0$, the operator $T(\cdot)$ is said to be $L$-Lipschitz if it satisfies

$$\|T(\mathbf{x}) - T(\mathbf{y})\|_2 \leq L\|\mathbf{x} - \mathbf{y}\|_2, \forall \, \mathbf{x}, \mathbf{y} \in \text{dom}(T). \tag{22}$$

• **Non-expansive operators:** An operator $T(\cdot)$ is said to be *non-expansive* if it satisfies

$$\|T(\mathbf{x}) - T(\mathbf{y})\|_2 \leq \|\mathbf{x} - \mathbf{y}\|_2, \forall \, \mathbf{x}, \mathbf{y} \in \text{dom}(T). \tag{23}$$

This implies that mapping a pair of points under a non-expansive operator does not increase the distance between them. Note that such an operator is 1-Lipschitz.

**Fact 1: (Beck, 2017, Theorem 5.4)** Let $\mathcal{X} \subset \mathbb{R}^n$ denote a non-empty, closed convex set and $\text{Proj}_{\mathcal{X}}(\mathbf{y})$ denote the Euclidean projection operator that projects a point $\mathbf{y}$ onto $\mathcal{X}$. Then, $\text{Proj}_{\mathcal{X}}(\cdot)$ is non-expansive.

• **Contractive operators:** An operator $T(\cdot)$ is said to be *contractive* if it satisfies

$$\|T(\mathbf{x}) - T(\mathbf{y})\|_2 < \|\mathbf{x} - \mathbf{y}\|_2, \forall \, \mathbf{x}, \mathbf{y} \in \text{dom}(T). \tag{24}$$

In other words, $T(\cdot)$ is $L$-Lipschitz with $L < 1$. In contrast with non-expansive operators, the distance between a pair of points strictly decreases under a contraction mapping.

**Fact 2: (Ryu & Yin, 2022, p. 33)** Let $T_1(\cdot)$ and $T_2(\cdot)$ denote a non-expansive and contraction operator, respectively. Then, their composition $T_1 \circ T_2$ is a contraction.

**Fact 3: [The Banach fixed-point theorem](Banach, 1922)** Let $T(\cdot)$ be a contractive mapping. Then, a fixed point of $T(\cdot)$ exists and it is unique.

## A.2. Euclidean projections onto the intersection of a hyperplane and a box constraint

Given a hyperplane $\mathcal{H}_{\mathbf{a},b} := \{\mathbf{x} : \mathbf{a}^\top \mathbf{x} = b\}$ where $\mathbf{a} \in \mathbb{R}^n \setminus \{\mathbf{0}\}$, $b \in \mathbb{R}$, and a box constraint of the form $\text{Box}[\boldsymbol{\ell}, \mathbf{u}] := \{\mathbf{x} : \boldsymbol{\ell} \leq \mathbf{x} \leq \mathbf{u}\}$, consider the general problem of computing the Euclidean projection of a point $\mathbf{y}$ onto the intersection $\mathcal{C} := \mathcal{H}_{\mathbf{a},b} \cap \text{Box}[\boldsymbol{\ell}, \mathbf{u}]$. This entails finding the minimizer of the following convex optimization problem

$$\text{Proj}_{\mathcal{C}}(\mathbf{y}) = \arg\min\left\{\|\mathbf{x} - \mathbf{y}\|_2^2 : \mathbf{a}^\top \mathbf{x} = b, \boldsymbol{\ell} \leq \mathbf{x} \leq \mathbf{u}\right\} \tag{25}$$

The following result provides a characterization of the solution.

**Theorem A.1.** *(Beck, 2017, Theorem 6.27) The solution of the projection problem* (25) *is*

$$\text{Proj}_{\mathcal{C}}(\mathbf{y}) = \text{Proj}_{\text{Box}[\boldsymbol{\ell},\mathbf{u}]}(\mathbf{y} - \lambda^* \mathbf{a}), \tag{26}$$

*where* $\text{Proj}_{\text{Box}[\boldsymbol{\ell},\mathbf{u}]}(\mathbf{z}) := \min(\max(\mathbf{z}, \boldsymbol{\ell}), \mathbf{u})$ *and* $\lambda^*$ *is the solution of the equation*

$$\mathbf{a}^\top \text{Proj}_{\text{Box}[\boldsymbol{\ell},\mathbf{u}]}(\mathbf{y} - \lambda \mathbf{a}) = b. \tag{27}$$

# B. Proofs

## B.1. Objective Function $f(\cdot)$

Recall the objective function

$$f(\mathbf{x}) := \frac{1}{2}(1-\gamma)\mathbf{x}^\top \mathbf{\Pi}^{-1}\mathbf{L}_{\mathrm{rw}}\mathbf{x} + \frac{1}{2}\gamma\|\mathbf{\Pi}^{-\frac{1}{2}}(\mathbf{x}-\mathbf{v})\|^2. \tag{28}$$

**Theorem B.1.** *If the graph is undirected or directed and time-reversible, then the objective function $f(\cdot)$ is strongly convex.*

*Proof.* Let $\mathbf{P}$ denote the probability transition matrix of a Markov chain with stationary distribution $\boldsymbol{\pi}$ obeying $\mathbf{P}\boldsymbol{\pi} = \boldsymbol{\pi}$. The Markov chain is said to be time-reversible (Levin & Peres, 2017) if the following balance equations are satisfied

$$\pi_j P_{ij} = \pi_i P_{ji}, \forall\, (i,j) \in [n] \times [n], \tag{29}$$

which can be represented in matrix form as

$$\mathbf{P}^\top = \mathbf{\Pi}^{-1}\mathbf{P}\mathbf{\Pi}. \tag{30}$$

For undirected graphs, the transition matrix $\mathbf{P} = \mathbf{A}\mathbf{D}^{-1}$ is time-reversible by construction. In the directed case, this need not apply, and hence the assumption. The implication of time-reversibility is that the matrix $\mathbf{S} := \mathbf{\Pi}^{-1}\mathbf{L}_{\mathrm{rw}}$ is symmetric, since

$$\mathbf{S}^\top = (\mathbf{\Pi}^{-1}\mathbf{L}_{\mathrm{rw}})^\top = \mathbf{L}_{\mathrm{rw}}^\top\mathbf{\Pi}^{-1} = (\mathbf{I} - \mathbf{P}^\top)\mathbf{\Pi}^{-1} = (\mathbf{I} - \mathbf{\Pi}^{-1}\mathbf{P}\mathbf{\Pi})\mathbf{\Pi}^{-1} = \mathbf{\Pi}^{-1} - \mathbf{\Pi}^{-1}\mathbf{P} = \mathbf{\Pi}^{-1}(\mathbf{I} - \mathbf{P}) \tag{31}$$

$$= \mathbf{\Pi}^{-1}\mathbf{L}_{\mathrm{rw}} = \mathbf{S}. \tag{32}$$

Hence, $\mathbf{S}$ has real eigen-values. Moreover, $\mathbf{S}$ can be shown to be positive semi-definite (denoted as $\mathbf{S} \succeq \mathbf{0}$). First, we express $\mathbf{S}$ as

$$\mathbf{S} = \mathbf{\Pi}^{-1}(\mathbf{I} - \mathbf{P}) = \mathbf{\Pi}^{-1/2}(\mathbf{I} - \mathbf{\Pi}^{-1/2}\mathbf{P}\mathbf{\Pi}^{1/2})\mathbf{\Pi}^{-1/2} \tag{33}$$

Note that $\bar{\mathbf{P}} := \mathbf{\Pi}^{-1/2}\mathbf{P}\mathbf{\Pi}^{1/2}$ is obtained by applying a similarity transformation on $\mathbf{P}$, and hence they possess the same eigen-values. Additionally, time-reversibility of $\mathbf{P}$ also implies that $\bar{\mathbf{P}}$ is symmetric. Since the largest eigen-value of $\mathbf{P}$ is $\lambda_{\max}(\mathbf{P}) = 1$, it follows that $\lambda_{\max}(\bar{\mathbf{P}}) = 1$. Hence, $1 - \lambda_{\max}(\bar{\mathbf{P}}) = 0$ is the smallest eigen-value of $\mathbf{I} - \bar{\mathbf{P}}$, from which it follows that $\mathbf{I} - \bar{\mathbf{P}}$ is positive semi-definite. Finally, $\mathbf{\Pi}$ is a diagonal matrix with strictly positive entries. Hence $\mathbf{S} = \mathbf{\Pi}^{-1/2}(\mathbf{I} - \bar{\mathbf{P}})\mathbf{\Pi}^{-1/2}$ is also positive semi-definite.

To conclude the proof, the first and second derivatives of $f(\cdot)$ are the following:

$$\nabla f(\mathbf{x}) = (1-\gamma)\mathbf{\Pi}^{-1}\mathbf{L}_{\mathrm{rw}}\mathbf{x} + \gamma\mathbf{\Pi}^{-1}(\mathbf{x}-\mathbf{v}), \tag{34}$$

$$\nabla^2 f(\mathbf{x}) = (1-\gamma)\mathbf{\Pi}^{-1}\mathbf{L}_{\mathrm{rw}} + \gamma\mathbf{\Pi}^{-1}. \tag{35}$$

Since $\gamma \in (0,1)$, the matrices $\mathbf{\Pi}^{-1}\mathbf{L}_{\mathrm{rw}} \succeq \mathbf{0}$ and $\mathbf{\Pi}^{-1} \succ 0$, it follows that $\nabla^2 f(\mathbf{x}) \succ 0$. We conclude that $f(\cdot)$ is strongly convex. $\square$

**Theorem B.2.** *The solution of $\min\{f(\mathbf{x})\}$ is equivalent to the original PageRank solution, $\mathbf{p}_o$.*

*Proof.* Since, $f(\cdot)$ is strongly convex, the problem $\min\{f(\mathbf{x})\}$ has a unique solution given by:

$$\nabla f(\mathbf{x}^*) = 0 \Leftrightarrow (1-\gamma)\mathbf{\Pi}^{-1}\mathbf{L}_{\mathrm{rw}}\mathbf{x}^* + \gamma\mathbf{\Pi}^{-1}(\mathbf{x}^* - \mathbf{v}) = 0 \tag{36}$$

$$\Leftrightarrow (1-\gamma)\mathbf{L}_{\mathrm{rw}}\mathbf{x}^* + \gamma(\mathbf{x}^* - \mathbf{v}) = 0 \tag{37}$$

$$\Leftrightarrow (1-\gamma)[\mathbf{I} - \mathbf{P}]\mathbf{x}^* + \gamma(\mathbf{x}^* - \mathbf{v}) = 0 \tag{38}$$

$$\Leftrightarrow \boxed{\mathbf{x}^* = \gamma[\mathbf{I} - (1-\gamma)\mathbf{P}]^{-1}\mathbf{v} = \mathbf{p}_o}. \tag{39}$$

$\square$

## B.2. Relation to Prior Variational Formulation of PageRank

In this appendix, we clarify the relation between our variational formulation of PageRank (4) and that proposed by Fountoulakis et al. (2019).

The work of Fountoulakis et al. (2019) introduced the following strongly convex quadratic cost function

$$F(\bar{\mathbf{x}}) = \frac{1}{2}\bar{\mathbf{x}}^\top \mathbf{Q}\bar{\mathbf{x}} - \alpha \mathbf{v}^\top \mathbf{D}^{-\frac{1}{2}}\bar{\mathbf{x}}, \tag{40}$$

where $\mathbf{Q} := \mathbf{D}^{-\frac{1}{2}}\left\{\mathbf{D} - \frac{1-\alpha}{2}(\mathbf{D}+\mathbf{A})\right\}\mathbf{D}^{-\frac{1}{2}}$. Setting the derivative $\nabla F(\bar{\mathbf{x}}) = 0$ and applying the change of co-ordinates $\bar{\mathbf{x}} = \mathbf{D}^{-\frac{1}{2}}\mathbf{x}$, we obtain the following linear system

$$\mathbf{x} = (1-\alpha)\mathbf{W}\mathbf{x} + \alpha\mathbf{v}, \tag{41}$$

where $\mathbf{W} = (\mathbf{I} + \mathbf{A}\mathbf{D}^{-1})/2 = (\mathbf{I} + \mathbf{P})/2$ is the lazy random walk matrix, and $\alpha \in (0,1)$ is the teleportation probability. The solution of this linear system also yields a PageRank vector w.r.t. the matrix $\mathbf{W}$.

For given teleportation vector $\mathbf{v}$ and probability $\alpha$, it turns out that this linear system is equivalent to solving the original PageRank system with the standard random walk matrix $\mathbf{P}$

$$\mathbf{x} = (1-\gamma)\mathbf{P}\mathbf{x} + \gamma\mathbf{v}, \tag{42}$$

if we just set the teleportation probability $\gamma = \frac{2\alpha}{1+\alpha}$ (see Proposition 3 in Andersen et al. (2006)). Hence, the solutions of these two PageRank systems are equivalent up to a scaling of the teleportation probabilities.

That being said, the variational formulation of Fountoulakis et al. (2019) is restricted to the setting of undirected graphs and is derived through analysis of the lazy random walk. In contrast, our variational formulation is based on the standard (non-lazy) random walk, and applies to both directed and undirected graphs.

An additional distinction concerns the coordinate system in which the optimization problem is expressed. In Fountoulakis et al. (2019), deriving the PageRank linear system (41) from (40) requires applying a change of coordinates to the PageRank vector $\mathbf{x}$, while the teleportation vector $\mathbf{v}$ remains unchanged. However, in order to recover a label spreading interpretation, an additional change of coordinates must be applied to the teleportation vector as well, namely $\bar{\mathbf{v}} = \mathbf{D}^{-\frac{1}{2}}\mathbf{v}$. Without this change, the solution does not admit a direct interpretation as label spreading. Under this change of co-ordinates, the linear system in (41) becomes

$$\begin{aligned}
\bar{\mathbf{x}} &= \frac{1-\alpha}{1+\alpha}\mathbf{D}^{-\frac{1}{2}}\mathbf{A}\mathbf{D}^{-\frac{1}{2}}\bar{\mathbf{x}} + \frac{2\alpha}{1+\alpha}\bar{\mathbf{v}} \\
\bar{\mathbf{x}} &= \frac{1-\alpha}{1+\alpha}\tilde{\mathbf{A}}\bar{\mathbf{x}} + \frac{2\alpha}{1+\alpha}\bar{\mathbf{v}}
\end{aligned} \tag{43}$$

with $\tilde{\mathbf{A}} := \mathbf{D}^{-\frac{1}{2}}\mathbf{A}\mathbf{D}^{-\frac{1}{2}}$ denoting the normalized adjacency matrix. This system can be solved via the following fixed point iterations

$$\bar{\mathbf{x}}^{(t+1)} = \frac{1-\alpha}{1+\alpha}\tilde{\mathbf{A}}\bar{\mathbf{x}}^{(t)} + \frac{2\alpha}{1+\alpha}\bar{\mathbf{v}}, \tag{44}$$

which are equivalent to our label spreading formulation (3) with parameter choice again $\gamma = \frac{2\alpha}{1+\alpha}$.

## B.3. Proof of Theorem 3.2

The proof comprises two parts. First, we assume that a fixed point $\mathbf{x}^*$ of the iterations exists and establish a geometric rate of convergence of the iterate sequence $\{\mathbf{x}^{(t)}\}_{t\geq 0}$ to $\mathbf{x}^*$. In the second part, we establish the existence and uniqueness of $\mathbf{x}^*$.

• **Step 1:** We equivalently express the iterations (6) as the following fixed point mapping

$$\mathbf{x}^{(t+1)} = \text{Proj}_{\mathcal{X}}((1-\gamma)\mathbf{P}\mathbf{x}^{(t)} + \gamma\mathbf{v}). \tag{45}$$

Let $\mathbf{x}^*$ be a fixed of the above mapping. By construction, $\mathbf{x}^*$ satisfies

$$\mathbf{x}^* = \text{Proj}_{\mathcal{X}}((1-\gamma)\mathbf{P}\mathbf{x}^* + \gamma\mathbf{v}). \tag{46}$$

Then, the $\ell_2$-distance between the iterate $\mathbf{x}^{(t+1)}$ and $\mathbf{x}^*$ (i.e., the distance to optimality) can be bounded as

$$
\begin{aligned}
\|\mathbf{x}^{(t+1)} - \mathbf{x}^*\|_2 &= \|\text{Proj}_{\mathcal{X}}((1-\gamma)\mathbf{P}\mathbf{x}^{(t)} + \gamma\mathbf{v}) - \text{Proj}_{\mathcal{X}}((1-\gamma)\mathbf{P}\mathbf{x}^* + \gamma\mathbf{v})\|_2 \\
&\leq \|(1-\gamma)\mathbf{P}(\mathbf{x}^{(t)} - \mathbf{x}^*)\|_2 \\
&\leq (1-\gamma)\|\mathbf{P}\|_2\|\mathbf{x}^{(t)} - \mathbf{x}^*\|_2 \\
&\leq (1-\gamma)\|\mathbf{x}^{(t)} - \mathbf{x}^*\|_2.
\end{aligned}
\tag{47}
$$

The second inequality follows from the non-expansive property of the projection operator, and the last inequality holds since $\mathbf{P}$ is a column-stochastic matrix; i.e., $\|\mathbf{P}\|_2 = \sigma_{\max}(\mathbf{P}) = 1$.

Applying the error bound (47) recursively, we obtain

$$
\|\mathbf{x}^{(t+1)} - \mathbf{x}^*\|_2 \leq (1-\gamma)^{t+1}\|\mathbf{x}^{(0)} - \mathbf{x}^*\|_2,
\tag{48}
$$

which shows that the distance between $\mathbf{x}^{(t)}$ and $\mathbf{x}^*$ vanishes to zero at a geometric rate of $(1-\gamma)^t$. We conclude that the sequence $\{\mathbf{x}^{(t)}\}_{t\geq 0}$ converges to $\mathbf{x}^*$.

• **Step 2:** To show existence and uniqueness of $\mathbf{x}^*$, it suffices to establish that the operator $T(\mathbf{x}) = \text{Proj}_{\mathcal{X}}((1-\gamma)\mathbf{P}\mathbf{x}^{(t)} + \gamma\mathbf{v})$ is contractive. Let $T_1(\mathbf{x}) := \text{Proj}_{\mathcal{X}}(\mathbf{x})$ and $T_2(\mathbf{x}) := (1-\gamma)\mathbf{P}\mathbf{x} + \gamma\mathbf{v}$. Hence, we can express $T$ as the composition of $T_1 \circ T_2$. Note that $T_1(\mathbf{x})$ is non-expansive, while $T_2(\mathbf{x})$ is $(1-\gamma)$-Lipschitz, and hence, contractive. This follows since $\|T_2(\mathbf{x}) - T_2(\mathbf{y})\|_2 \leq (1-\gamma)\|\mathbf{P}\|_2\|\mathbf{x} - \mathbf{y}\|_2 = (1-\gamma)\|\mathbf{x} - \mathbf{y}\|_2$. We conclude that $T(\cdot)$ is contractive, being the composition of a non-expansive and a contractive operator. Hence, by the Banach fixed-point theorem (Banach, 1922), $T(\cdot)$ has a unique fixed point $\mathbf{x}^*$.

### B.4. Proof of Theorem 3.3

Consider the constrained optimization problem (5):

$$
\min_{\mathbf{x}\in\mathcal{X}} f(\mathbf{x}).
\tag{49}
$$

Since $f(\cdot)$ is strongly convex (Theorem B.1) and $\mathcal{X}$ is a non-empty, closed convex set, it possesses a unique solution $\mathbf{x}^*$. By definition, $\mathbf{x}^*$ satisfies the following first-order optimality condition (Boyd & Vandenberghe, 2004, Section 4.2.3)

$$
\nabla f(\mathbf{x}^*)^\top(\mathbf{y} - \mathbf{x}^*) \geq 0, \forall\, \mathbf{y} \in \mathcal{X}.
\tag{50}
$$

The gradient of $f(\cdot)$ is given by Eq. (34):

$$
\begin{aligned}
\nabla f(\mathbf{x}) &= (1-\gamma)\mathbf{\Pi}^{-1}\mathbf{L}_{\text{rw}}\mathbf{x} + \gamma\mathbf{\Pi}^{-1}(\mathbf{x} - \mathbf{v}) \\
&= \mathbf{\Pi}^{-1}([\mathbf{I} - (1-\gamma)\mathbf{P}]\mathbf{x} - \gamma\mathbf{v}) \\
\Rightarrow \mathbf{\Pi}\cdot\nabla f(\mathbf{x}) &= [\mathbf{I} - (1-\gamma)\mathbf{P}]\mathbf{x} - \gamma\mathbf{v}.
\end{aligned}
\tag{51}
$$

Define $r(\mathbf{x}) := [\mathbf{I} - (1-\gamma)\mathbf{P}]\mathbf{x} - \gamma\mathbf{v}$. This allows us to compactly express (51) as $\nabla f(\mathbf{x}) = \mathbf{\Pi}^{-1}r(\mathbf{x})$. Then, the optimality condition (59) can be equivalently expressed w.r.t. $\mathbf{x}^*$ as

$$
r(\mathbf{x}^*)^\top\mathbf{\Pi}^{-1}(\mathbf{y} - \mathbf{x}^*) \geq 0, \forall\, \mathbf{y} \in \mathcal{X}.
\tag{52}
$$

Meanwhile, let $\hat{\mathbf{x}}$ denote the fixed point of the iterations (6). Hence, it satisfies the condition

$$
\hat{\mathbf{x}} = \text{Proj}_{\mathcal{X}}((1-\gamma)\mathbf{P}\hat{\mathbf{x}} + \gamma\mathbf{v}) = \text{Proj}_{\mathcal{X}}(\hat{\mathbf{x}} - r(\hat{\mathbf{x}})).
\tag{53}
$$

We conclude that

$$
\hat{\mathbf{x}} = \arg\min_{\mathbf{x}\in\mathcal{X}} \|\mathbf{x} - (\hat{\mathbf{x}} - r(\hat{\mathbf{x}}))\|_2^2
\tag{54}
$$

Since the above problem is convex, applying the first-order optimality condition implies that $\hat{\mathbf{x}}$ must satisfy

$$
\begin{aligned}
&(\hat{\mathbf{x}} - (\hat{\mathbf{x}} - r(\hat{\mathbf{x}})))^\top(\mathbf{y} - \hat{\mathbf{x}}) \geq 0, \forall\, \mathbf{y} \in \mathcal{X} \\
\Leftrightarrow\; &r(\hat{\mathbf{x}})^\top(\mathbf{y} - \hat{\mathbf{x}}) \geq 0, \forall\, \mathbf{y} \in \mathcal{X}.
\end{aligned}
\tag{55}
$$

It remains to show that $\hat{\mathbf{x}}$ satisfying (55) also satisfies (52). This follows from the fact that

$$
\begin{aligned}
r(\hat{\mathbf{x}})^\top \mathbf{\Pi}^{-1}(\mathbf{y} - \hat{\mathbf{x}}) &= \sum_{i=1}^n \frac{1}{\pi_i} r(\hat{\mathbf{x}})_i (\mathbf{y} - \hat{\mathbf{x}})_i \\
&\geq \sum_{i=1}^n r(\hat{\mathbf{x}})_i (\mathbf{y} - \hat{\mathbf{x}})_i \\
&= r(\hat{\mathbf{x}})^\top (\mathbf{y} - \hat{\mathbf{x}}), \forall\, \mathbf{y} \in \mathcal{X}
\end{aligned}
\tag{56}
$$

The inequality holds from the fact that $\mathbf{\Pi} = \mathrm{diag}(\boldsymbol{\pi})$, where $\boldsymbol{\pi}$ is a probability vector, hence $\pi_i \in (0, 1], \forall\, i \in \{1, \ldots, n\}$. Since we assume that the graph is connected, we have that $\pi_i \neq 0$.
Consequently, we obtain the desired implication

$$
\begin{aligned}
&r(\hat{\mathbf{x}})^\top (\mathbf{y} - \hat{\mathbf{x}}) \geq 0, \forall\, \mathbf{y} \in \mathcal{X} \\
&\implies r(\hat{\mathbf{x}})^\top \mathbf{\Pi}^{-1}(\mathbf{y} - \hat{\mathbf{x}}), \forall\, \mathbf{y} \in \mathcal{X}.
\end{aligned}
\tag{57}
$$

Since the solution of (5) is unique and $\hat{\mathbf{x}}$ is a solution (5), then it corresponds to its unique solution. Hence, the unique minimizer $\mathbf{x}^*$ of (5) is the fixed-point of the FairRARI iterations (6).

### B.5. Time-reversibility in the directed setting

Consider a directed graph $\mathcal{G}$ whose transition matrix $\mathbf{P}$ is not time-reversible. In this case, the claim of Theorem 3.3 does not apply. In this section, we will make the weaker assumption that the Markov chain $\tilde{\mathbf{P}} = (1 - \gamma)\mathbf{P} + \gamma(1/n)\mathbf{e}\mathbf{e}^\top$ is time-reversible. This is a more justifiable assumption, as the incorporation of teleportation ensures that the necessary condition for balance in (29) is satisfied; i.e., $\tilde{P}_{ij} > 0 \Leftrightarrow \tilde{P}_{ji} > 0$. Moreover, the transition matrix $(1/n) \cdot \mathbf{e}\mathbf{e}^\top$ is time-reversible by design. Hence, as $\gamma \to 1$, $\tilde{\mathbf{P}}$ better approximates a time-reversible Markov chain. Additionally, in App. G.2, we performed the time-reversibility test for $\tilde{\mathbf{P}}$ on the directed graph datasets that we used, and discovered that they very closely approximate this property, even for $\gamma = 0.15$.

Using the fact that $\tilde{\mathbf{P}}$ is time-reversible, we will use the random walk Laplacian $\tilde{\mathbf{L}}_{\mathrm{rw}} := \mathbf{I} - \tilde{\mathbf{P}}$ to derive an analogue of Theorem 3.3. Consider the quadratic function

$$
\tilde{f}(\mathbf{x}) = \frac{1}{2}\mathbf{x}^\top \mathbf{\Pi}^{-1} \tilde{\mathbf{L}}_{\mathrm{rw}} \mathbf{x},
\tag{58}
$$

where $\mathbf{\Pi} = \mathrm{diag}(\boldsymbol{\pi})$, and $\boldsymbol{\pi}$ is the stationary distribution of $\tilde{\mathbf{P}}$, obeying $\tilde{\mathbf{P}}\boldsymbol{\pi} = \boldsymbol{\pi}$. Since $\tilde{\mathbf{P}}$ is time-reversible, by a similar line of reasoning as in the proof of Theorem B.1, it holds that $\mathbf{\Pi}^{-1}\tilde{\mathbf{L}}_{\mathrm{rw}}$ is symmetric and positive semi-definite. Hence, the quadratic $\tilde{f}(\cdot)$ is a convex function. Our main result is the following analogue of Theorem 3.3.

**Theorem B.3.** *Fix a directed graph $\mathcal{G}$. If $\tilde{\mathbf{P}}$ is a time-reversible Markov chain, then the fixed point of the FairRARI iterations (6) is a minimizer of the constrained convex optimization problem $\min_{\mathbf{x} \in \mathcal{X}} \tilde{f}(\mathbf{x})$, where $\mathcal{X} \subset \mathbb{R}^n$ is a closed convex set.*

*Proof.* Since $\mathcal{X}$ is a non-empty, closed convex set, the constrained optimization problem $\min_{\mathbf{x} \in \mathcal{X}} \tilde{f}(\mathbf{x})$ possesses a minimizer $\mathbf{x}^*$. By definition, $\mathbf{x}^*$ satisfies the following first-order optimality condition (Boyd & Vandenberghe, 2004, Section 4.2.3)

$$
\nabla \tilde{f}(\mathbf{x}^*)^\top (\mathbf{y} - \mathbf{x}^*) \geq 0, \forall\, \mathbf{y} \in \mathcal{X},
\tag{59}
$$

with the gradient of $\tilde{f}(\cdot)$

$$
\begin{aligned}
\nabla \tilde{f}(\mathbf{x}) &= \mathbf{\Pi}^{-1}\tilde{\mathbf{L}}_{\mathrm{rw}}\mathbf{x} \\
&= \mathbf{\Pi}^{-1}\left([\mathbf{I} - (1 - \gamma)\mathbf{P}]\mathbf{x} - \gamma \cdot \mathbf{v}\right) \\
\Rightarrow \mathbf{\Pi} \cdot \nabla \tilde{f}(\mathbf{x}) &= [\mathbf{I} - (1 - \gamma)\mathbf{P}]\mathbf{x} - \gamma\mathbf{v},
\end{aligned}
\tag{60}
$$

being the same as that of the cost function $f(\cdot)$ in (5) (with $\mathbf{v} = \frac{1}{n}\mathbf{e}$). By repeating the arguments of the proof of Theorem 3.3 verbatim (after equation (51) and onward), the desired claim follows. □

### B.6. Proof of Theorem 5.1

*Proof.* The proof comprises two parts. First, we are going to show that Post-Processing produced results that are agnostic to the graph structure, and the show that for FairRARI they are not. We will make use of the fact that the constraint set $\mathcal{X}$ is convex.

● **Post-Processing:** Since $\mathcal{X}$ is convex, the optimization problem that the Post-Processing method is solving, comprises the following inequality and equality constraints:

$$
\begin{aligned}
\min_{\mathbf{x}} \quad & \|\mathbf{x} - \mathbf{p}_\mathrm{o}\|_2^2 \\
\text{s.t.} \quad & w_i(\mathbf{x}) \le 0, \ i = 1, \dots, p \\
& u_j(\mathbf{x}) = 0, \ j = 1, \dots, q
\end{aligned}
\tag{61}
$$

From the stationarity KKT condition we get that

$$
\nabla \mathcal{L}(\mathbf{x}^*, \boldsymbol{\lambda}, \boldsymbol{\mu}) = 0 \Leftrightarrow \nabla\{\|\mathbf{x}^* - \mathbf{p}_\mathrm{o}\|_2^2\} + \sum_{i=1}^{p} \lambda_i \nabla w_i(\mathbf{x}^*) + \sum_{j=1}^{q} \mu_j \nabla u_j(\mathbf{x}^*) = 0
\tag{62}
$$

$$
2(\mathbf{x}^* - \mathbf{p}_\mathrm{o}) + \sum_{i=1}^{p} \lambda_i \nabla w_i(\mathbf{x}^*) + \sum_{j=1}^{q} \mu_j \nabla u_j(\mathbf{x}^*) = 0
\tag{63}
$$

$$
\Leftrightarrow \mathbf{x}^* = \mathbf{p}_\mathrm{o} - \frac{1}{2}\left( \sum_{i=1}^{p} \lambda_i \nabla w_i(\mathbf{x}^*) + \sum_{j=1}^{q} \mu_j \nabla u_j(\mathbf{x}^*) \right),
\tag{64}
$$

where $\mathbf{p}_\mathrm{o} = \gamma[\mathbf{I} - (1 - \gamma)\mathbf{P}]^{-1}\mathbf{v}$ is the original PR vector.
From the result above we can see that the Post-Processing method is agnostic to the graph structure.

● **FairRARI:** We have shown that FairRARI is solving the following constrained, strongly convex QP

$$
\min \{ f(\mathbf{x}) \text{ s.t. } \mathbf{x} \in \mathcal{X} \},
\tag{65}
$$

where

$$
f(\mathbf{x}) := \frac{1}{2}(1 - \gamma)\mathbf{x}^\top \boldsymbol{\Pi}^{-1} \mathbf{L}_\mathrm{rw}\mathbf{x} + \frac{1}{2}\gamma\|\boldsymbol{\Pi}^{-\frac{1}{2}}(\mathbf{x} - \mathbf{v})\|^2,
\tag{66}
$$

with

$$
\nabla f(\mathbf{x}) = \boldsymbol{\Pi}^{-1}\left([\mathbf{I} - (1 - \gamma)\mathbf{P}]\mathbf{x} - \gamma\mathbf{v}\right).
\tag{67}
$$

Since $\mathcal{X}$ is convex, the optimization problem comprises of the following inequality and equality constraints:

$$
\begin{aligned}
\min_{\mathbf{x}} \quad & f(\mathbf{x}) \\
\text{s.t.} \quad & w_i(\mathbf{x}) \le 0, \ i = 1, \dots, p \\
& u_j(\mathbf{x}) = 0, \ j = 1, \dots, q
\end{aligned}
\tag{68}
$$

From the stationarity KKT condition we get that

$$\nabla \mathcal{L}(\mathbf{x}^*, \boldsymbol{\lambda}, \boldsymbol{\mu}) = 0 \Leftrightarrow \nabla f(\mathbf{x}^*) + \sum_{i=1}^{p} \lambda_i \nabla w_i(\mathbf{x}^*) + \sum_{j=1}^{q} \mu_j \nabla u_j(\mathbf{x}^*) = 0 \tag{69}$$

$$\mathbf{\Pi}^{-1} \left( [\mathbf{I} - (1-\gamma)\mathbf{P}]\mathbf{x}^* - \gamma \mathbf{v} \right) + \sum_{i=1}^{p} \lambda_i \nabla w_i(\mathbf{x}^*) + \sum_{j=1}^{q} \mu_j \nabla u_j(\mathbf{x}^*) = 0 \tag{70}$$

$$\left( [\mathbf{I} - (1-\gamma)\mathbf{P}]\mathbf{x}^* - \gamma \mathbf{v} \right) + \mathbf{\Pi} \sum_{i=1}^{p} \lambda_i \nabla w_i(\mathbf{x}^*) + \mathbf{\Pi} \sum_{j=1}^{q} \mu_j \nabla u_j(\mathbf{x}^*) = 0 \tag{71}$$

$$\mathbf{x}^* = \gamma[\mathbf{I} - (1-\gamma)\mathbf{P}]^{-1}\mathbf{v} - [\mathbf{I} - (1-\gamma)\mathbf{P}]^{-1}\mathbf{\Pi} \left( \sum_{i=1}^{p} \lambda_i \nabla w_i(\mathbf{x}^*) + \sum_{j=1}^{q} \mu_j \nabla u_j(\mathbf{x}^*) \right) \tag{72}$$

$$\Leftrightarrow \mathbf{x}^* = \mathbf{p}_\mathrm{o} - [\mathbf{I} - (1-\gamma)\mathbf{P}]^{-1}\mathbf{\Pi} \left( \sum_{i=1}^{p} \lambda_i \nabla w_i(\mathbf{x}^*) + \sum_{j=1}^{q} \mu_j \nabla u_j(\mathbf{x}^*) \right), \tag{73}$$

where $\mathbf{p}_\mathrm{o} = \gamma[\mathbf{I} - (1-\gamma)\mathbf{P}]^{-1}\mathbf{v}$ is the original PR vector.
From the result above we can see that FaiRARI's solution makes use of the graph structure since it is dependent on its stationary distribution $\mathbf{\Pi}$. $\qquad \square$

### B.7. Proof of Corollary 5.2

*Proof.* We consider the case of $\phi$-fairness for two groups, $\mathcal{S}_1$ and $\mathcal{S}_2$. Since $\mathcal{X} = \{\mathbf{x} \geq 0 \; : \; \mathbf{e}_{\mathcal{S}_k}^{\top}\mathbf{x} = \phi_k, k = 1, 2\}$ is convex, with $\phi_1 = \phi$ and $\phi_2 = 1 - \phi$, the optimization problem that the Post-Processing method is solving is the following

$$\min_{\mathbf{x} \in \mathcal{X}} \quad \|\mathbf{x} - \mathbf{p}_\mathrm{o}\|_2^2 \tag{74}$$

with $\mathbf{p}_\mathrm{o}$ the solution of the original PageRank, and $\mathbf{e}_{\mathcal{S}_1}^{\top}\mathbf{x} = \phi_\mathrm{o}$ and $\mathbf{e}_{\mathcal{S}_2}^{\top}\mathbf{x} = 1 - \phi_\mathrm{o}$ The solution of this problem, as it is given by Theorem 4.1 is

$$\left\{ \begin{array}{c} \mathbf{x}_{\mathcal{S}_k} = \max\{0, \mathbf{y}_{\mathcal{S}_k} - \lambda_k^* \mathbf{e}_{\mathcal{S}_k}\} \\ \sum_{i \in \mathcal{S}_k} \max\{0, y_i - \lambda_k^*\} = \phi_k \end{array} \right\} \quad k = 1, 2. \tag{75}$$

W.l.o.g. we assume that $\phi_1 - \phi_\mathrm{o} = \epsilon \geq 0$, i.e., $\phi_1 = \phi_\mathrm{o} + \epsilon$ and $\phi_2 = 1 - \phi_\mathrm{o} - \epsilon$.

Hence, for $\mathcal{S}_1$ we get

$$\phi_1 = \phi_\mathrm{o} + \epsilon = \sum_{i \in \mathcal{S}_1} \max\{0, \mathbf{p}_\mathrm{o}(i) - \lambda_1^*\} = \sum_{i \in \mathcal{S}_1} \{\mathbf{p}_\mathrm{o}(i) - \lambda_1^*\} = \sum_{i \in \mathcal{S}_1} \mathbf{p}_\mathrm{o}(i) - \sum_{i \in \mathcal{S}_1} \lambda_1^* = \phi_\mathrm{o} - |\mathcal{S}_1|\lambda_1^* \Rightarrow \lambda_1^* = -\frac{\epsilon}{|\mathcal{S}_1|}. \tag{76}$$

We remove the max operator after the second equation because $\sum_{i \in \mathcal{S}_1} \mathbf{p}_\mathrm{o}(i) = \phi_\mathrm{o} \leq \phi_\mathrm{o} + \epsilon$. This implies that the sum needs to be increased. Since the same $\lambda_1^*$ applies to all $i$, each $\mathbf{p}_\mathrm{o}(i)$ will be increased by the same amount, resulting in values that are strictly greater than zero.

Let's now split $\mathcal{S}_2$ as follows:

$$\mathcal{S}_2 = \mathcal{S}_2^+ \cup \mathcal{S}_2^-, \quad \mathcal{S}_2^+ = \{i \in \mathcal{S}_2 : \mathbf{p}_\mathrm{o}(i) \geq \lambda_2^*\}, \quad \mathcal{S}_2^- = \{i \in \mathcal{S}_2 : \mathbf{p}_\mathrm{o}(i) < \lambda_2^*\}. \tag{77}$$

Hence,

$$\phi_2 = 1 - \phi_\mathrm{o} - \epsilon = \sum_{i \in \mathcal{S}_2} \max\{0, \mathbf{p}_\mathrm{o}(i) - \lambda_2^*\} = \sum_{i \in \mathcal{S}_2^+} \{\mathbf{p}_\mathrm{o}(i) - \lambda_2^*\} = \sum_{i \in \mathcal{S}_2^+} \mathbf{p}_\mathrm{o}(i) - \sum_{i \in \mathcal{S}_2^+} \lambda_2^* = 1 - \phi_\mathrm{o} - c - |\mathcal{S}_2^+|\lambda_2^* \tag{78}$$

$$\Rightarrow \lambda_2^* = \frac{\epsilon - c}{|\mathcal{S}_2^+|}, \tag{79}$$

with $\sum_{i \in \mathcal{S}_2} \mathbf{p}_o(i) = 1 - \phi_o$, $\sum_{i \in \mathcal{S}_2^+} \mathbf{p}_o(i) = 1 - \phi_o - c$ and $c = \sum_{i \in \mathcal{S}_2^-} \mathbf{p}_o(i)$. Resulting to

$$\mathbf{x}^*(i) = \begin{cases} \mathbf{p}_o(i) + \frac{\epsilon}{|\mathcal{S}_1|}, & i \in \mathcal{S}_1 \\ \mathbf{p}_o(i) - \frac{\epsilon - c}{|\mathcal{S}_2^+|}, & i \in \mathcal{S}_2^+ \\ 0, & i \in \mathcal{S}_2^- \end{cases}. \tag{80}$$

□

*Proof for FairRARI (Extra).* This time we consider the general $\phi$-sum-fairness problem, for $K > 2$. FairRARI is solving the following constrained, strongly convex QP

$$\min_{\mathbf{x} \in \Delta_\phi(\mathcal{V})} f(\mathbf{x}). \tag{81}$$

The Lagrangian is

$$\mathcal{L}(\mathbf{x}, \boldsymbol{\lambda}, \boldsymbol{\mu}) = f(\mathbf{x}) - \boldsymbol{\lambda}^\top \mathbf{x} + \boldsymbol{\mu}^\top (\mathbf{E}\mathbf{x} - \boldsymbol{\phi}). \tag{82}$$

where $\mathbf{E}$ is the $k \times n$ matrix that has $\mathbf{e}_{\mathcal{S}_k}^\top, \forall k \in [K]$, as rows.
From the stationarity KKT condition we get that

$$\nabla \mathcal{L}(\mathbf{x}^*, \boldsymbol{\lambda}, \boldsymbol{\mu}) = 0 \Leftrightarrow \nabla f(\mathbf{x}^*) - \boldsymbol{\lambda} + \mathbf{E}^\top \boldsymbol{\mu} = 0 \tag{83}$$

$$\Leftrightarrow \mathbf{x}^* = [\mathbf{I} - (1 - \gamma)\mathbf{P}]^{-1}\mathbf{\Pi}(\gamma\mathbf{\Pi}^{-1}\mathbf{v} + \boldsymbol{\lambda} - \mathbf{E}^\top \boldsymbol{\mu}) \tag{84}$$

$$\Leftrightarrow \mathbf{x}^* = \gamma[\mathbf{I} - (1 - \gamma)\mathbf{P}]^{-1}\mathbf{v} + [\mathbf{I} - (1 - \gamma)\mathbf{P}]^{-1}\mathbf{\Pi}(\boldsymbol{\lambda} - \mathbf{E}^\top \boldsymbol{\mu}) \tag{85}$$

$$\Leftrightarrow \mathbf{x}^* = \mathbf{p}_o + [\mathbf{I} - (1 - \gamma)\mathbf{P}]^{-1}\mathbf{\Pi}(\boldsymbol{\lambda} - \mathbf{E}^\top \boldsymbol{\mu}). \tag{86}$$

Also, it holds that

$$\mathbf{e}_{\mathcal{S}_1}^\top \mathbf{x}^* = \phi_o + \epsilon \Leftrightarrow \mathbf{e}_{\mathcal{S}_1}^\top [\mathbf{I} - (1 - \gamma)\mathbf{P}]^{-1}\mathbf{\Pi}(\boldsymbol{\lambda} - \mathbf{E}^\top \boldsymbol{\mu}) = \epsilon, \tag{87}$$

and

$$\mathbf{e}_{\mathcal{S}_2}^\top \mathbf{x}^* = 1 - \phi_o - \epsilon \Leftrightarrow \mathbf{e}_{\mathcal{S}_2}^\top [\mathbf{I} - (1 - \gamma)\mathbf{P}]^{-1}\mathbf{\Pi}(\boldsymbol{\lambda} - \mathbf{E}^\top \boldsymbol{\mu}) = -\epsilon. \tag{88}$$

□

## C. Efficient Fair Projections

### C.1. $\phi$-sum-fairness

#### C.1.1. PROOF OF THEOREM 4.1

*Proof.* We have the following set

$$\Delta_\phi(\mathcal{V}) := \{\mathbf{x} \geq 0 : \mathbf{e}_{\mathcal{S_k}}^\top \mathbf{x} = \phi_k, \forall k = [K]\}, \tag{89}$$

Based on Theorem A.1, and the fact that the constraints are decomposable, since $\{\mathcal{S}_k\}_{k=1}^K$ is a partition of $\mathcal{V}$, the Euclidean projection of a vector $\mathbf{y}$ on this set is

$$\left[P_{\Delta_\phi(\mathcal{V})}(\mathbf{y})\right]_i = \left[P_{\text{Box}[0,\infty]}(\mathbf{y}_{\mathcal{S}_k} - \lambda_k^* \mathbf{e}_{\mathcal{S}_k})\right]_i, \ i \in \mathcal{S}_k, \forall k \in [K], \tag{90}$$

where the variables $\{\lambda_k^*\}_{k=1}^K$ satisfy

$$\mathbf{e}_{\mathcal{S}_k}^\top P_{\text{Box}[0,\infty]}(\mathbf{y}_{\mathcal{S}_k} - \lambda_k^* \mathbf{e}_{\mathcal{S}_k}) = \phi_k, \forall k \in [K] \tag{91}$$

Hence, each element of the projection is

$$x_i = \left[P_{\Delta_\phi}(\mathbf{y})\right]_i = \left[P_{\text{Box}[0,\infty]}(\mathbf{y}_{\mathcal{S}_k} - \lambda_k^* \mathbf{e}_{\mathcal{S}_k})\right]_i = \max\{0, y_i - \lambda_k^*\}, \ i \in \mathcal{S}_k, \forall k \in [K] \tag{92}$$

or equivalently

$$\mathbf{x}_{\mathcal{S}_k} = \max\{0, \mathbf{y}_{\mathcal{S}_k} - \lambda_k^* \mathbf{e}_{\mathcal{S}_k}\}, \forall k \in [K] \tag{93}$$

with

$$\mathbf{e}_{\mathcal{S}_k}^\top P_{\text{Box}[0,\infty]}(\mathbf{y}_{\mathcal{S}_k} - \lambda_k^* \mathbf{e}_{\mathcal{S}_k}) = \phi_k \Leftrightarrow \sum_{i \in \mathcal{S}_k} \max\{0, y_i - \lambda_k^*\} = \phi_k, \forall k \in [K]. \tag{94}$$

□

C.1.2. SOLVING FOR THE DUAL VARIABLES $\lambda_k^*$

For a fixed vector $\mathbf{y}$, define

$$g_k(\lambda_k) := \sum_{i \in \mathcal{S}_k} \max\{0, y_i - \lambda_k\} - \phi_k, \forall\, k \in [K]. \tag{95}$$

It is easy to see that each optimal dual variable $\lambda_k^*$ corresponds to the root of the non-smooth, non-linear equation $g_k(\lambda_k^*) = 0$. Since each function $g_k(\cdot)$ is continuous and monotonically non-increasing w.r.t $\lambda_k$ (Beck, 2017), the root can be determined via a bisection-search procedure. In order to initialize the procedure, we need to determine lower and upper limits $\lambda_k^{\min}$ and $\lambda_k^{\max}$ respectively such that $\lambda_k^{\min} < \lambda_k^{\max}$, $g_k(\lambda_k^{\min}) > 0$ and $g_k(\lambda_k^{\max}) < 0$. A suitable choice is

$$\lambda_k^{\min} = \min_{i \in \mathcal{S}_k} y_i - \frac{\phi_k}{n_k}, \quad \lambda_k^{\max} = \max_{i \in \mathcal{S}_k} y_i. \tag{96}$$

It can be seen that $0 < g_k(\lambda_k^{\min}) < n_k - 1$ and $g_k(\lambda_k^{\max}) = -\phi_k < 0$.

***Complexity:*** For a prescribed exit tolerance $\epsilon_k$, the number of bisection iterations required is $O(\log_2(\lambda_k^{\max} - \lambda_k^{\min})/\epsilon_k) = O(\log_2(1/\epsilon_k))$. Since each iteration entails $O(n_k)$ complexity, the overall procedure costs $O(n_k \cdot \log_2(1/\epsilon_k))$. Hence, for a fixed tolerance $\epsilon_k$, the complexity is linear in $n_k$. Since we consider $K$ groups of vertices, hence each Fair Projection requires a total of $K$ bisections. These bisections can be performed either sequentially or in parallel. For sequential execution, with fixed tolerance values $\{\epsilon_k\}_{k=1}^K$, the total complexity is $\sum_{k=1}^K O(n_k) = O(n)$. In contrast, for parallel execution, the complexity is determined by the largest group, i.e., $O(n_k)$, where $n_k = \max_{i \in [K]} n_i$.

## C.2. $\alpha$-min-fairness

C.2.1. PROOF OF THEOREM 4.2

*Proof.* We have the following set

$$\Delta^{\boldsymbol{\alpha}}(\mathcal{A}) := \{\mathbf{x} \geq 0 \;:\; \mathbf{e}^\top \mathbf{x} = 1, \; \min_{i \in \mathcal{A}_k} x_i \geq \alpha_k, \; \forall\, k \in [K]\}, \tag{97}$$

Based on Theorem A.1, the Euclidean projection of a vector $\mathbf{y}$ on this set is

$$\big[\mathrm{P}_{\Delta^{\boldsymbol{\alpha}}(\mathcal{A})}(\mathbf{y})\big]_i = \big[\mathrm{P}_{\mathrm{Box}[\boldsymbol{\ell}, \infty]}(\mathbf{y} - \lambda^* \mathbf{e})\big]_i, \; i \in \mathcal{V}, \tag{98}$$

where the variable $\lambda^*$ satisfies

$$\mathbf{e}^\top \mathrm{P}_{\mathrm{Box}[\boldsymbol{\ell}, \infty]}(\mathbf{y} - \lambda^* \mathbf{e}) = 1, \tag{99}$$

and $\boldsymbol{\ell}$ is the following vector

$$\ell_i = \begin{cases} \alpha_k, & i \in \mathcal{A}_k \\ 0, & i \in \bar{\mathcal{A}}_k \end{cases}, \; \forall\, k \in [K]. \tag{100}$$

Hence, each element of the projection is

$$x_i = \big[\mathrm{P}_{\mathrm{Box}[\boldsymbol{\ell}, \infty]}(\mathbf{y} - \lambda^* \mathbf{e})\big]_i = \begin{cases} \max\{\alpha_k, y_i - \lambda^*\}, & i \in \mathcal{A}_k \\ \max\{0, y_i - \lambda^*\}, & i \in \bar{\mathcal{A}}_k \end{cases}, \; \forall\, k \in [K]. \tag{101}$$

with

$$\mathbf{e}^\top \mathrm{P}_{\mathrm{Box}[\boldsymbol{\ell}, \infty]}(\mathbf{y} - \lambda^* \mathbf{e}) = 1 \Leftrightarrow \sum_{k=1}^K \left\{ \sum_{i \in \mathcal{A}_k} \max\{\alpha_k, y_i - \lambda^*\} + \sum_{i \in \bar{\mathcal{A}}_k} \max\{0, y_i - \lambda^*\} \right\} = 1. \tag{102}$$

$\square$

C.2.2. SOLVING FOR THE DUAL VARIABLE $\lambda^*$

For a fixed vector $\mathbf{y}$, define

$$h(\lambda) := \sum_{k=1}^K \left\{ \sum_{i \in \mathcal{A}_k} \max\{\alpha_k, y_i - \lambda\} + \sum_{i \in \bar{\mathcal{A}}_k} \max\{0, y_i - \lambda\} \right\} - 1. \tag{103}$$

Similarly to the $\phi$-sum-fairness case, it is easy to see that the function $h(\cdot)$ is continuous and monotonically non-increasing w.r.t $\lambda$, with $\lambda^{\min}$ and $\lambda^{\max}$ such that $\lambda^{\min} < \lambda^{\max}$, $h(\lambda^{\min}) > 0$ and $h(\lambda^{\max}) < 0$. A suitable choice is

$$\lambda^{\min} = \min_{i \in \mathcal{V}} y_i - \frac{1}{n}, \ \lambda^{\max} = \max_{i \in \mathcal{V}} y_i. \tag{104}$$

It can be seen that $h(\lambda^{\min}) > 0$ and $h(\lambda^{\max}) < 0$.

***Complexity:*** Following the previous analysis, for a fixed tolerance $\epsilon$, the complexity of the Fair Projections is linear to the number of vertices of the graph, i.e., $O(n)$, since for the $\alpha$-min-fairness case we need to perform only one bisection per projection.

### C.3. $\phi$-sum + $\alpha$-min-fairness

C.3.1. PROOF OF THEOREM 4.3

*Proof.* We have the following set

$$\Delta_{\phi}^{\alpha}(\mathcal{V}, \mathcal{A}) := \{\mathbf{x} \geq 0 : \ \mathbf{e}_{\mathcal{S}_{\mathbf{k}}}^{\top} \mathbf{x} = \phi_k, \ \min_{i \in \mathcal{A}_k} x_i \geq \alpha_k, \ \forall \, k \in [K]\}, \tag{105}$$

Based on Theorem A.1, and the fact that the constraints are decomposable, since $\{\mathcal{S}_k\}_{k=1}^{K}$ is a partition of $\mathcal{V}$, the Euclidean projection of a vector $\mathbf{y}$ on this set is

$$\left[\mathrm{P}_{\Delta_{\phi}^{\alpha}(\mathcal{V}, \mathcal{A})}(\mathbf{y})\right]_i = \left[\mathrm{P}_{\mathrm{Box}[\boldsymbol{\ell}, \infty]}(\mathbf{y} - \lambda_k^* \mathbf{e}_{\mathcal{S}_k})\right]_i, \ i \in \mathcal{S}_k, \ \forall \, k \in [K], \tag{106}$$

where the variables $\{\lambda_k^*\}_{k=1}^{K}$ satisfy

$$\mathbf{e}_{\mathcal{S}_k}^{\top} \mathrm{P}_{\mathrm{Box}[\boldsymbol{\ell}, \infty]}(\mathbf{y}_{\mathcal{S}_k} - \lambda_k^* \mathbf{e}_{\mathcal{S}_k}) = \phi_k, \ \forall \, k \in [K], \tag{107}$$

and $\boldsymbol{\ell}$ the following vector

$$\ell_i = \begin{cases} \alpha_k, & i \in \mathcal{A}_k \\ 0, & i \in \bar{\mathcal{A}}_k \end{cases}, \ \forall \, k \in [K]. \tag{108}$$

Hence, each element of the projection is

$$x_i = \left[\mathrm{P}_{\mathrm{Box}[\boldsymbol{\ell}, \infty]}(\mathbf{y}_{\mathcal{S}_k} - \lambda_k^* \mathbf{e}_{\mathcal{S}_k})\right]_i = \begin{cases} \max\{\alpha_k, y_i - \lambda_k^*\}, & i \in \mathcal{A}_k \\ \max\{0, y_i - \lambda_k^*\}, & i \in \bar{\mathcal{A}}_k \end{cases}, \ \forall \, k \in [K] \tag{109}$$

with

$$\mathbf{e}_{\mathcal{S}_k}^{\top} \mathrm{P}_{\mathrm{Box}[\boldsymbol{\ell}, \infty]}(\mathbf{y} - \lambda_k^* \mathbf{e}_{\mathcal{S}_k}) = \phi_k \Leftrightarrow \sum_{i \in \mathcal{A}_k} \max\{\alpha_k, y_i - \lambda_k^*\} + \sum_{i \in \bar{\mathcal{A}}_k} \max\{0, y_i - \lambda_k^*\} = \phi_k, \ \forall \, k \in [K]. \tag{110}$$

$\square$

C.3.2. SOLVING FOR THE DUAL VARIABLES $\lambda_k^*$

For a fixed vector $\mathbf{y}$, define

$$u_k(\lambda_k) := \sum_{i \in \mathcal{A}_k} \max\{\alpha_k, y_i - \lambda_k\} + \sum_{i \in \bar{\mathcal{A}}_k} \max\{0, y_i - \lambda_k\} - \phi_k, \tag{111}$$

for all $k \in [K]$. Similarly to the previous cases, each function $u_k(\cdot)$ is continuous and monotonically non-increasing w.r.t $\lambda_k$, with $\lambda_k^{\min}$ and $\lambda_k^{\max}$ such that $\lambda_k^{\min} < \lambda_k^{\max}$, $v_k(\lambda_k^{\min}) > 0$ and $v_k(\lambda_k^{\max}) < 0$, implying that bisection-search can be applied to solve for the root of each equation $u_k(\lambda_k) = 0, \forall \, k \in [K]$. A suitable choice is

$$\lambda_k^{\min} = \min_{i \in \mathcal{S}_k} y_i - \frac{\phi_k}{n_k}, \ \lambda_k^{\max} = \max_{i \in \mathcal{S}_k} y_i. \tag{112}$$

It can be seen that $v_k(\lambda_k^{\min}) > 0$ and $v_k(\lambda_k^{\max}) < 0$.

***Complexity:*** The complexity of the Fair Projections in this case is identical to that of the $\phi$-sum-fairness, i.e., $O(n)$ for sequential execution, and $O(n_k)$, with $n_k = \max_{i \in [K]} n_i$, for parallel.

# D. Algorithms

---

**Algorithm 2** FAIR-PROJECTION

---

**Input:** $\mathbf{y}$, $\mathcal{X}$, $\mathcal{G} = (\mathcal{V}, \mathcal{E})$
**Output:** Fair Projection outcome
{The following loop can be performed either sequentially or in parallel}
**for** $k \in 1, \ldots, K$ **do**
$\quad \lambda_k^{\max} \leftarrow \max_{i \in \mathcal{S}_k} y_i$
$\quad$**if** $\mathcal{X} = \Delta_\phi(\mathcal{V})$ **then**
$\quad\quad \lambda_k^{\min} \leftarrow \min_{i \in \mathcal{S}_k} y_i - \frac{\phi_k}{n_k}$
$\quad\quad \lambda_k^* \leftarrow \text{BISECTION}(g_k, \lambda_k^{\min}, \lambda_k^{\max}, \epsilon)$
$\quad\quad x_i^* \leftarrow \max\{0, y_i - \lambda_k^*\}, \ i \in \mathcal{S}_k$
$\quad$**else if** $\mathcal{X} = \Delta^{\boldsymbol{\alpha}}(\mathcal{A})$ **then**
$\quad\quad \lambda^{\min} \leftarrow \min_{i \in \mathcal{V}} y_i - \frac{1}{n}, \lambda^{\max} \leftarrow \max_{i \in \mathcal{V}} y_i$
$\quad\quad \lambda^* \leftarrow \text{BISECTION}(h, \lambda^{\min}, \lambda^{\max}, \epsilon)$
$\quad\quad x_i^* \leftarrow \begin{cases} \max\{\alpha_k, y_i - \lambda^*\}, & i \in \mathcal{A}_k \\ \max\{0, y_i - \lambda^*\}, & i \notin \mathcal{A}_k \end{cases}$
$\quad$**else if** $\mathcal{X} = \Delta_{\boldsymbol{\phi}}^{\boldsymbol{\alpha}}(\mathcal{V}, \mathcal{A})$ **then**
$\quad\quad \lambda_k^{\min} \leftarrow \min_{i \in \mathcal{S}_k} y_i - \frac{\phi_k}{n_k}$
$\quad\quad \lambda_k^* \leftarrow \text{BISECTION}(v_k, \lambda_k^{\min}, \lambda_k^{\max}, \epsilon)$
$\quad\quad x_i^* \leftarrow \begin{cases} \max\{\alpha_k, y_i - \lambda_k^*\}, & i \in \mathcal{A}_k \\ \max\{0, y_i - \lambda_k^*\}, & i \notin \mathcal{A}_k \end{cases}, \ i \in \mathcal{S}_k$
$\quad$**end if**
**end for**
**return** $\mathbf{x}^*$

---

**Algorithm 3** BISECTION

---

**Input:** $f$, $\lambda^{\min}$, $\lambda^{\max}$, $\epsilon$
**Output:** Optimal $\lambda$ value
**Initialize:** $\lambda_u \leftarrow \lambda^{\max}, \lambda_l \leftarrow \lambda^{\min}$
**while** $f(\lambda_l) - f(\lambda_u) \geq \epsilon$ **do**
$\quad \lambda_m \leftarrow (\lambda_l + \lambda_u)/2$
$\quad$**if** $f(\lambda_m) < 0$ **then**
$\quad\quad \lambda_u \leftarrow \lambda_m$
$\quad$**else**
$\quad\quad \lambda_l \leftarrow \lambda_m$
$\quad$**end if**
**end while**
**return** $\lambda_m$

---

# E. Datasets

A summary of the statistics of all the datasets used in our experiments can be found in Tab. 1.

| Dataset | | $n$ | $m$ | $K$ | $n_i$ | $n_i/n$ | $\phi_o$ |
|---|---|---|---|---|---|---|---|
| PolBooks | UnDirected | 92 | 374 | 2 | $(43, 49)$ | $(0.47, 0.53)$ | $(0.47, 0.53)$ |
| PolBlogs | Directed | 1222 | 16717 | 2 | $(586, 636)$ | $(0.48, 0.52)$ | $(0.25, 0.75)$ |
| Erdos | UnDirected | 6927 | 8472 | 2 | $(57, 6870)$ | $(0.01, 0.99)$ | $(0.01, 0.99)$ |
| TwitchPTBR-2 | UnDirected | 1912 | 31299 | 2 | $(661, 1251)$ | $(0.35, 0.65)$ | $(0.34, 0.66)$ |
| TwitchRU-2 | UnDirected | 4385 | 37304 | 2 | $(1075, 3310)$ | $(0.25, 0.75)$ | $(0.25, 0.75)$ |
| TwitchES-2 | UnDirected | 4648 | 59382 | 2 | $(1360, 3288)$ | $(0.29, 0.71)$ | $(0.30, 0.70)$ |
| TwitchFR-2 | UnDirected | 6549 | 112666 | 2 | $(2414, 4135)$ | $(0.37, 0.63)$ | $(0.36, 0.64)$ |
| TwitchENGB-2 | UnDirected | 7126 | 35324 | 2 | $(3888, 3238)$ | $(0.55, 0.45)$ | $(0.56, 0.44)$ |
| TwitchDE-2 | UnDirected | 9498 | 153138 | 2 | $(5742, 3756)$ | $(0.60, 0.40)$ | $(0.60, 0.40)$ |
| DBLP-G | UnDirected | 16501 | 66613 | 2 | $(4242, 12259)$ | $(0.26, 0.74)$ | $(0.25, 0.75)$ |
| DBLP-P | UnDirected | 16501 | 66613 | 2 | $(1325, 15176)$ | $(0.08, 0.92)$ | $(0.08, 0.92)$ |
| Twitter | UnDirected | 18470 | 48053 | 2 | $(11355, 7115)$ | $(0.61, 0.39)$ | $(0.61, 0.39)$ |
| Deezer | UnDirected | 28281 | 92752 | 2 | $(12538, 15743)$ | $(0.44, 0.56)$ | $(0.44, 0.56)$ |
| Github | UnDirected | 37700 | 289003 | 2 | $(9739, 27961)$ | $(0.26, 0.74)$ | $(0.25, 0.75)$ |
| Slashdot | Directed | 82168 | 948464 | 2 | $(20543, 61625)$ | $(0.25, 0.75)$ | $(0.27, 0.73)$ |
| TwitchGAMERS | UnDirected | 168114 | 6797557 | 2 | $(79033, 89081)$ | $(0.47, 0.53)$ | $(0.47, 0.53)$ |
| TwitchPTBR | UnDirected | 1912 | 31299 | 4 | $(560, 101, 178, 1073)$ | $(0.29, 0.05, 0.10, 0.56)$ | $(0.29, 0.05, 0.11, 0.55)$ |
| TwitchRU | UnDirected | 4385 | 37304 | 4 | $(956, 119, 355, 2955)$ | $(0.22, 0.03, 0.08, 0.67)$ | $(0.22, 0.03, 0.08, 0.67)$ |
| TwitchES | UnDirected | 4648 | 59382 | 4 | $(1253, 107, 288, 3000)$ | $(0.27, 0.02, 0.06, 0.65)$ | $(0.27, 0.02, 0.07, 0.64)$ |
| TwitchFR | UnDirected | 6549 | 112666 | 4 | $(2255, 159, 249, 3886)$ | $(0.34, 0.02, 0.04, 0.60)$ | $(0.34, 0.02, 0.04, 0.60)$ |
| TwitchENGB | UnDirected | 7126 | 35324 | 4 | $(3701, 187, 197, 3041)$ | $(0.52, 0.03, 0.03, 0.42)$ | $(0.54, 0.02, 0.03, 0.41)$ |
| TwitchDE | UnDirected | 9498 | 153138 | 4 | $(5380, 362, 235, 3521)$ | $(0.57, 0.04, 0.02, 0.37)$ | $(0.58, 0.03, 0.02, 0.37)$ |

*Table 1.* Full summary of dataset statistics: number of vertices $(n)$, number of edges $(m)$, number of groups $(K)$, size of each group $(n_i)$, group-wise vertex proportions in the graph $(n_i/n)$, and fairness values of original PR $(\phi_o)$

**Political Books Dataset (PolBooks)** (Adamic & Glance, 2005): The vertices of the network are books on US politics included in the Amazon catalog, with an edge connecting two books if they are frequently co-purchased by the same buyers[3]. Each book is categorized based on its political stance, with possible labels including liberal, neutral, and conservative. In some cases we use a 2 group alternative of this dataset, the POLBOOKS (2), where we focused solely on the subgraph formed by liberal and conservative books, resulting in 92 vertices connected by a total of 374 edges, as curated by Anagnostopoulos et al. (2024; 2020).

**Political Blogs Dataset (PolBlogs)** (Adamic & Glance, 2005): The vertices of the network are weblogs on US politics, with edges representing hyperlinks. Each blog is categorized by its political stance, left or right, with the left ones composing the protected set.

**Paul Erdős co-authorship network (Erdos)** (Rossi & Ahmed, 2015): This is the co-authorship graph around Paul Erdős. The network is as of 2002, and contains people who have, directly and indirectly, written papers with Paul Erdős. This network is used to define the "Erdős number", i.e., the distance between any vertex and Paul Erdős. The attribute associated with each author is their gender. This attribute is being created by the names of the authors using the help of ChatGPT.

**DBLP Gender co-authorship network (DBLP-G)** (Tsioutsiouliklis et al., 2022): This co-authorship network is constructed from DBLP and includes a subset of data mining and database conferences spanning the years 2011 to 2020. The attribute associated with each author is their gender, which is inferred using the Python package, *gender-guesser*[4], as curated by Tsioutsiouliklis et al. (2022).

**DBLP Publication co-authorship network (DBLP-P)** (Tsioutsiouliklis et al., 2022): The same network as DBLP-G but with the attribute associated with each author being their if they are newcomers, i.e., if their first publication is in 2016 or later, as curated by Tsioutsiouliklis et al. (2022).

**Twitter Retweet Political Network (Twitter)** (Rossi & Ahmed, 2015; Conover et al., 2011): The vertices are Twitter users,

---

[3] https://websites.umich.edu/~mejn/netdata/
[4] https://pypi.org/project/gender-guesser/

and the edges represent whether the users have retweeted each other. The attribute associated with each user is their political orientation (Tsioutsiouliklis et al., 2021).

**Deezer Europe Social Network (Deezer)** (Rozemberczki & Sarkar, 2020): The vertices represent users of Deezer from European countries, and the edges are mutual follower connections among them. The attribute associated with each user is their gender.

**GitHub Developers (GitHub)** (Rozemberczki et al., 2021): The vertices represent developers in GitHub who have starred at least 10 repositories, and the edges are mutual follower relationships between them. The attribute associated with each user is whether they are web or a machine learning developers.

**Slashdot Users Network (Slashdot)** (Leskovec et al., 2009): A directed network representing friend/foe relationships between users on Slashdot. Since this dataset lacks group labels, the authors of (Wang et al., 2026) apply the METIS graph partitioning algorithm (Karypis & Kumar, 1997) to divide the vertices into groups.

**Twitch Users Social Networks (TwitchPTBR/RU/ES/FR/ENGB/DE)** (Rozemberczki et al., 2021): The vertices represent users of Twitch that stream in a certain language (PTBR/RU/ES/FR/ENGB/DE), and the edges are mutual friendships between them. These datasets are being used with either 2 or 4 groups. For the 2 group case the attribute associated with each user is if the user is using mature language. For the 4 group case we split each on of the previous groups into 2 more using the "partner" feature of the dataset.

**Twitch Gamers Social Networks (TwitchGAMERS)** (Rozemberczki & Sarkar, 2021): The vertices represent users of Twitch collected from the public API in Spring 2018, and the edges are mutual follower between them. The attribute associated with each user is again if the user is using mature language.

## F. Baselines

In this section we describe the details of the baselines used in our experiments.

▷ In the work of Tsioutsiouliklis et al. (2021), they consider the case of having just 2 groups of vertices, the group $R$ (protected) and the group $B$ (unprotected). Hence, they achieve $\phi$-sum-fairness when the sum of the scores in group $R$ is equal to $\phi$ and in $B$ equal to $1 - \phi$.

● FSPR: Considering $\mathbf{p}_o$ as the Original PageRank solution, $\mathbf{Q} = \gamma[\mathbf{I} - (1 - \gamma)\mathbf{P}]^{-1}$, and $\mathbf{Q}_R$ the vector that is the sum of the columns of $\mathbf{Q}$ in the set $R$, they define the following optimization problem

$$
\begin{aligned}
\min_{\mathbf{x}} \quad & \|\mathbf{x}^\top \mathbf{Q} - \mathbf{p}_o\| \\
\text{s.t.} \quad & \mathbf{x}^\top \mathbf{Q}_R = \phi \\
& \sum_{i=1}^n x_i = 1 \\
& 0 \le x_i \le 1, \ i = 1, \dots, n
\end{aligned}
\tag{113}
$$

● LFPR: For the LFPR family of algorithms they modify the transition matrix and the teleportation vector in order to achieve the desired fairness. So, the modified matrices and vectors are the following:

○ LFPR$_N$: Let $out_R(i)$ and $out_B(i)$ be the number of outgoing edges from vertex $i$ to red and blue vertices, respectively. They define $\mathbf{P}_R$ as the stochastic matrix that handles the transition to red vertices, or random jumps if such edges do not exist.

$$
\mathbf{P}_R[i,j] = \begin{cases} \frac{1}{out_R(i)} & \text{if } (i,j) \in \mathcal{E} \text{ and } j \in R \\ \frac{1}{|R|} & \text{if } out_R(i) = 0 \text{ and } j \in R \\ 0 & \text{otherwise} \end{cases}
\tag{114}
$$

The transition matrix $\mathbf{P}_B$ for the blue vertices is defined accordingly. Then, the final transition matrix $\mathbf{P}_N$ for LFPR$_N$ is:

$$
\mathbf{P}_N = \phi \mathbf{P}_R + (1 - \phi)\mathbf{P}_B.
\tag{115}
$$

Then, they also modify the teleportation vector $\mathbf{v}_N$, with $\mathbf{v}_N[i] = \frac{\phi}{|R|}$, if $i \in R$, and $\mathbf{v}_N[i] = \frac{\phi}{|B|}$, if $i \in B$. Hence, the LFPR$_N$ PageRank vector is

$$
\mathbf{p}_N^\top = (1 - \gamma)\mathbf{p}_N^\top \mathbf{P}_N + \gamma \mathbf{v}_N^\top.
\tag{116}
$$

$\circ$ LFPR$_\text{U}$: In this case we have the following:

$$\mathbf{P}_L[i,j] = \begin{cases} \frac{\phi}{out_R(i)} & \text{if } (i,j) \in \mathcal{E} \text{ and } j \in L_R \\ \frac{1-\phi}{out_B(i)} & \text{if } (i,j) \in \mathcal{E} \text{ and } j \in L_B \\ 0 & \text{otherwise} \end{cases}, \tag{117}$$

with the LFPR$_\text{U}$ PageRank vector being:

$$\mathbf{p}_L^\top = (1-\gamma)\mathbf{p}_L^\top(\mathbf{P}_L + \delta_R\mathbf{x}^\top + \delta_B\mathbf{y}^\top) + \gamma\mathbf{v}_N^\top, \tag{118}$$

with $\mathbf{x}[i] = \frac{1}{|R|}$, if $i \in R$, $\mathbf{y}[i] = \frac{1}{|B|}$, if $i \in B$, $\delta_R(i) = \phi - \frac{(1-\phi)\cdot out_R(i)}{out_B(i)}$, and $\delta_B(i) = (1-\phi) - \frac{\phi\cdot out_R(i)}{out_B(i)}$.

$\circ$ LFPR$_\text{P}$: This case is similar to the previous one, but with $\mathbf{x}[i] = \frac{\mathbf{p}_\circ[i]}{\sum_{i\in R}\mathbf{p}_\circ[i]}$, if $i \in R$, and $\mathbf{y}[i] = \frac{\mathbf{p}_\circ[i]}{\sum_{i\in B}\mathbf{p}_\circ[i]}$, if $i \in B$.

$\triangleright$ In a separate line of work, FairWalk (Rahman et al., 2019) and CrossWalk (Khajehnejad et al., 2022) are fairness-aware adaptations of node2vec (Grover & Leskovec, 2016) that modify the random walk process to ensure better representation of minority groups in graph embeddings.

$\bullet$ FairWalk: FairWalk can be seen as a variant of LFPR$_\text{N}$. We will present it in the case of two groups. Its teleportation vector being $\mathbf{v}_\text{FairWalk} = 0$ and the transition matrix for the red vertices is the following

$$\mathbf{P}_R[i,j] = \begin{cases} \frac{1}{out_R(i)} & \text{if } (i,j) \in \mathcal{E} \text{ and } j \in R \\ 0 & \text{otherwise} \end{cases} \tag{119}$$

Similarly, the transition matrix $\mathbf{P}_B$ for the blue vertices is defined accordingly. Then, the final transition matrix $\mathbf{P}_\text{FairWalk}$ for FairWalk is:

$$\mathbf{P}_\text{FairWalk} = \phi\mathbf{P}_R + (1-\phi)\mathbf{P}_B. \tag{120}$$

Not that $\mathbf{P}_R$ of FairWalk is the same as that of LFPR$_\text{N}$, but without the second branch.

$\bullet$ CrossWalk: CrossWalk retains the core ideas of FairWalk, with the addition of carefully re-weighting inner-group connections by assigning larger weights to edges closer to group peripheries.

$\triangleright$ Wang et al. (2026) propose an approach for incorporating group-fairness into the PageRank algorithm by reweighting the transition probabilities in the underlying transition matrix. They propose two different algorithms, FairGD and AdaptGD.

$\bullet$ FairGD: FairGD is the algorithm proposed to solve the following problem:

Given a set of target fairness levels $\{\phi_k\}_{k=1}^K$ satisfying $\phi_k \geq 0, \sum_{k=1}^K \phi_k = 1$, the sum of the PR scores for each group $k$ equals $\phi_k$, the goal is to find a new transition matrix $\mathbf{P}^*$ that minimizes the fairness-loss function $L(\mathbf{P}, \gamma, \mathbf{v}) = \frac{1}{K}\sum_{k=1}^K(\mathbf{e}_{\mathcal{S}_k}^\top\mathbf{p}_\circ - \phi_k)^2$, where $\mathbf{p}_\circ$ is the original PR vector given the parameters $(\mathbf{P}, \gamma, \mathbf{v})$, i.e.,

$$\mathbf{P}^* = \underset{\hat{\mathbf{P}}\in\mathcal{C}(\mathbf{P})}{\arg\min}\, L(\hat{\mathbf{P}}, \gamma, \mathbf{v}) \tag{121}$$

where

$$\mathcal{C}(\mathbf{P}) = \{\hat{\mathbf{P}} \in \mathbb{R}_{\geq 0}^{n\times n} \,|\, \hat{\mathbf{P}}\mathbf{1} = \mathbf{1}, \hat{\mathbf{P}}_{ij} = 0 \text{ if } \mathbf{P}_{ij} = 0\} \tag{122}$$

is the set of row-stochastic matrices that have non-negative entries, and the edge-reweighting of $\mathcal{G}$ can take place only on existing edges.

$\bullet$ AdaptGD: AdaptGD is a FairGD variant which enhances PageRank by considering random walks that (re)start at each group and by penalizing the fairness loss for each such (re)start. More specifically, let $\mathbf{v}_k = \frac{1}{|\mathcal{S}_k|}\mathbf{e}_{\mathcal{S}_k}$ be the vector indicating that the random walk restarts uniformly only from group $\mathcal{S}_k$. AdaptGD is proposed to solve the following problem:

Given a set of target fairness levels $\{\phi_k\}_{k=1}^K$ satisfying $\phi_k \geq 0, \sum_{k=1}^K \phi_k = 1$, the sum of the PR scores for each group $k$ equals $\phi_k$, the goal is to find a new transition matrix $\mathbf{P}^*$ that minimizes the *group-adapted* fairness-loss function

$$L_g(\mathbf{P}, \gamma) = \frac{1}{K}\sum_{\ell=1}^K L(\mathbf{P}, \gamma, \mathbf{v}_\ell) = \frac{1}{K^2}\sum_{\ell=1}^K\sum_{k=1}^K(\mathbf{e}_{\mathcal{S}_k}^\top\mathbf{p}_\ell - \phi_k)^2, \tag{123}$$

where $\mathbf{p}_\ell$ is the PR vector given the parameters $(\mathbf{P}, \gamma, \mathbf{v}_\ell)$, i.e.,

$$\mathbf{P}^* = \underset{\hat{\mathbf{P}} \in \mathcal{C}(\mathbf{P})}{\arg\min} L(\hat{\mathbf{P}}, \gamma). \tag{124}$$

• Restricted FairGD and Restricted AdaptGD: In addition, they propose some restricted versions of FairGD and AdaptGD, in which $\hat{\mathbf{P}}$ is constrained to $\hat{\mathbf{P}} \in \mathcal{C}_R(\mathbf{P}) \cap \mathcal{C}(\mathbf{P})$, where

$$\mathcal{C}_R(\mathbf{P}) = \{\hat{\mathbf{P}} \in \mathbb{R}_{\geq 0}^{n \times n} \,|\, (1 - \delta)\mathbf{P}[i, j] - \epsilon \leq \hat{\mathbf{P}}[i, j] \leq (1 + \delta)\mathbf{P}[i, j] + \epsilon\}, \tag{125}$$

for $\delta, \epsilon \geq 0$.

## G. Additional Experiments

### G.1. Experimental Setup

All experiments were performed on a single machine, with Intel i7-13700K CPU @ 3.4GHz and 128GB of main memory running Python 3.12.0 and C++ 11. The code of the baselines was obtained from the publicly available repositories associated with the respective publications. We based our implementation of FairRARI on the NetworkX[5] implementation of PageRank, in which we incorporate the Fair Projections. For the stopping criteria we set $10^4$ iterations or $\epsilon = n \cdot 10^{-6}$ (the same as NetworkX). Regarding the parameters of PageRank for all baselines, we set $\gamma = 0.15$ and $\mathbf{v}$ to be the uniform vector.

### G.2. Testing for Time-Reversibility

Given a directed graph $\mathcal{G}$, testing for time-reversibility boils down to checking the balance Equations (29), (30). A necessary condition is that if a directed arc from node $i$ to $j$ exists, a "reverse" arc from $j$ to $i$ must also exist. If this test fails for any node pair, we can conclude that time-reversibility does not hold. If this test passes, we can then proceed to check if the balance Equation (30) is satisfied. This can be done by computing the stationary distribution $\mathbf{P}\boldsymbol{\pi} = \boldsymbol{\pi}$ of the random walk matrix $\mathbf{P}$, setting $\boldsymbol{\Pi} = \mathrm{diag}(\boldsymbol{\pi})$, and then checking whether $\boldsymbol{\Pi}\mathbf{P}^\top = \mathbf{P}\boldsymbol{\Pi}$ is satisfied or not.

We applied the time-reversibility test on the random walk matrix $\mathbf{P}$ of the directed graphs used in our study. In the leftmost column of the tables below, we display the Frobenius norm error, Error $:= \|\boldsymbol{\Pi}\mathbf{P}^\top - \mathbf{P}\boldsymbol{\Pi}\|_F$, and the Mean Absolute Error (MAE) in the balance Equation (30). Although these results indicate that time-reversibility does not hold *exactly*, it is evident that $\mathbf{P}$ well approximates this property. We additionally tested for time-reversibility of the combined Markov chain $\tilde{\mathbf{P}} := (1 - \gamma)\mathbf{P} + \gamma\frac{1}{n}\mathbf{e}\mathbf{e}^T$ for increasing values of $\gamma$, and we depict the error metrics in the remaining columns of the tables below. The observed decrease in both metrics as $\gamma$ increases provides empirical confirmation that incorporating teleportation drives $\tilde{\mathbf{P}}$ toward time-reversibility. At the limit of $\gamma = 1$, $\tilde{\mathbf{P}} = \frac{1}{n}\mathbf{e}\mathbf{e}^T$, which corresponds to a fully time-reversible Markov chain.

|  | PolBlogs ($\gamma = 0$) | PolBlogs ($\gamma = 0.15$) | PolBlogs ($\gamma = 0.85$) |
|---|---|---|---|
| Error | 0.0183 | 0.0118 | 0.0019 |
| MAE | $10 \times 10^{-7}$ | $9 \times 10^{-7}$ | $2 \times 10^{-7}$ |

*Table 2.* Time-reversibility test on PolBlogs dataset for different $\gamma$ values.

|  | Slashdot ($\gamma = 0$) | Slashdot ($\gamma = 0.15$) | Slashdot ($\gamma = 0.85$) |
|---|---|---|---|
| Error | 0.0007 | 0.0006 | 0.0002 |
| MAE | $6 \times 10^{-11}$ | $4 \times 10^{-11}$ | $3 \times 10^{-11}$ |

*Table 3.* Time-reversibility test on Slashdot dataset for different $\gamma$ values.

---

[5] https://networkx.org/

## G.3. Wall-Clock Time

We evaluate the wall-clock time of FairRARI and the post-processing method, comparing them with prior methods. Each method and dataset is tested over 100 runs, reporting mean and standard deviation for each $\phi$. Since our implementation is in Python while the prior methods use C++, we do not directly compare the wall-clock times. Instead, we compare the percentage of increase in runtime of each respective Fair PR (FPR) implementation, relative to the original PR (OPR), i.e., $100 \cdot (t(\text{FPR}) - t(\text{oPR}))/t(\text{oPR})$. Fig. 6 shows that FairRARI does not significantly impact the time complexity of the OPR algorithm. In contrast, prior methods demonstrate considerable instability in terms of wall-clock time, with deviations in some experiments exceeding 5000. This can be a matter of implementation, but the take home message here is that FairRARI maintains the same complexity as the OPR. This is evident also in the Figs. 7 and 8, in which FairRARI can converge faster than the OPR. Note that for FairRARI we use the sequential and not the parallel implementation.

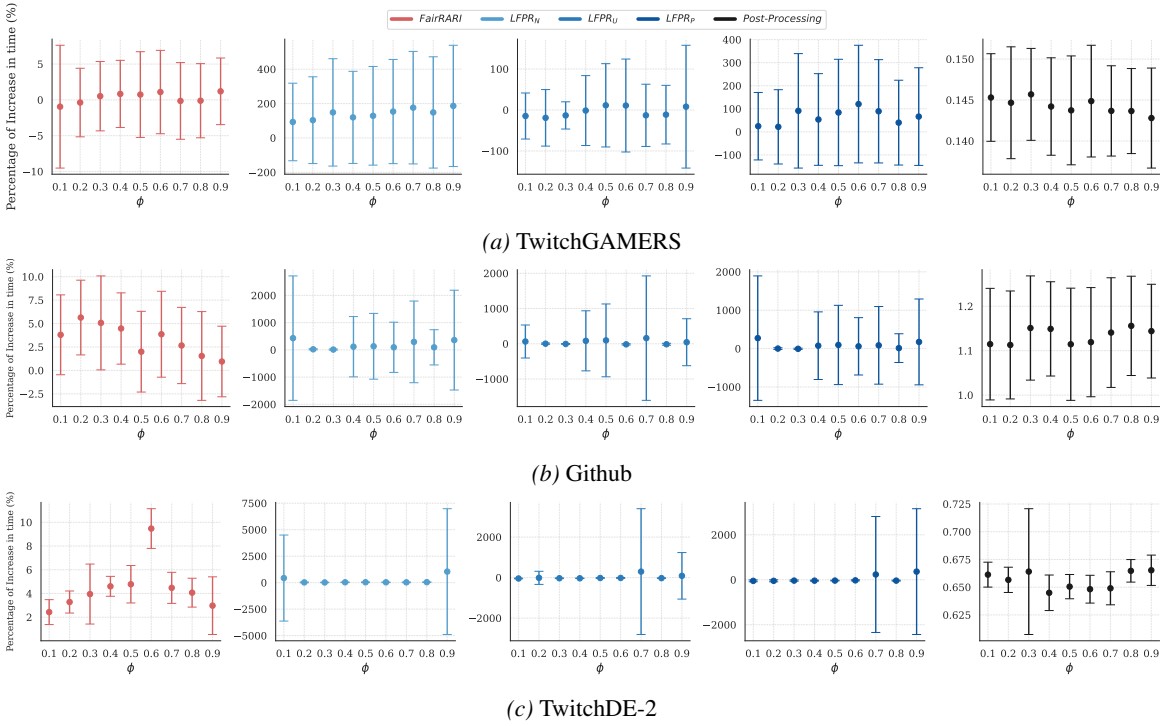

*Figure 6.* Percentage of increase in time over original PR for each one of the implementations, for the $\phi$-sum-fairness problem.

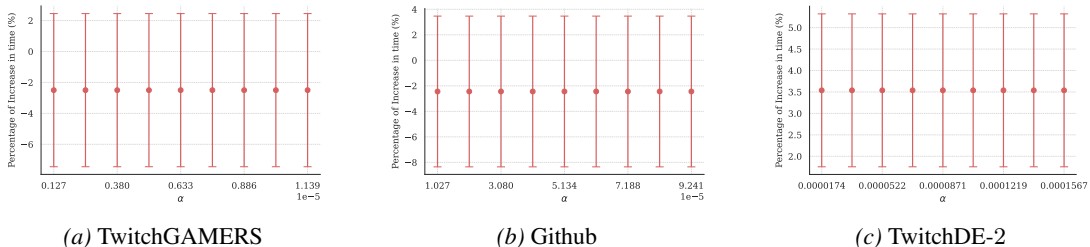

*Figure 7.* Percentage of increase in time over original PR for FairRARI, for the $\alpha$-min-fairness problem.

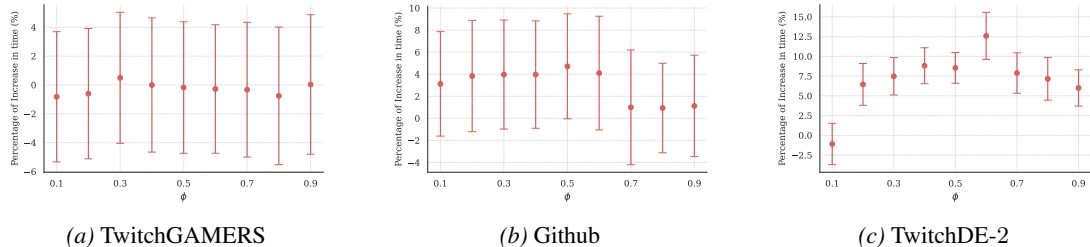

*(a)* TwitchGAMERS                    *(b)* Github                    *(c)* TwitchDE-2

*Figure 8.* Percentage of increase in time over original PR for FairRARI, for the $\phi$-sum + $\alpha$-min-fairness problem.

## G.4. $\phi$-sum-fairness

### G.4.1. TOTAL VARIATION

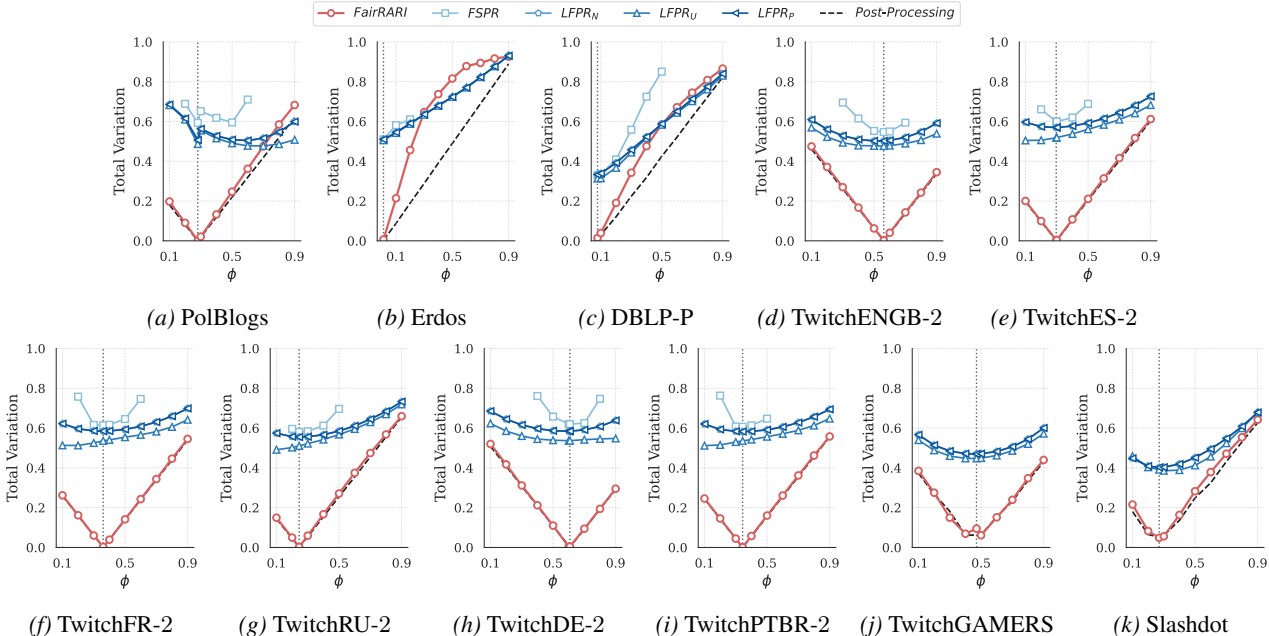

*Figure 9.* Extra experiments similar to Fig. 2. Comparing Fair PageRank solutions of prior methods and FairRARI (Ours), on the rest of datasets with 2 groups. It is evident that our algorithm outperforms the existing approaches. For various values of $\phi$, FairRARI consistently yields a significantly lower TV, very close the lower bound obtained via the post-processing method, indicating that the allocation of fair PR scores is much closer to that of the original PR.

### G.4.2. KENDALL TAU RANK CORRELATION COEFFICIENT

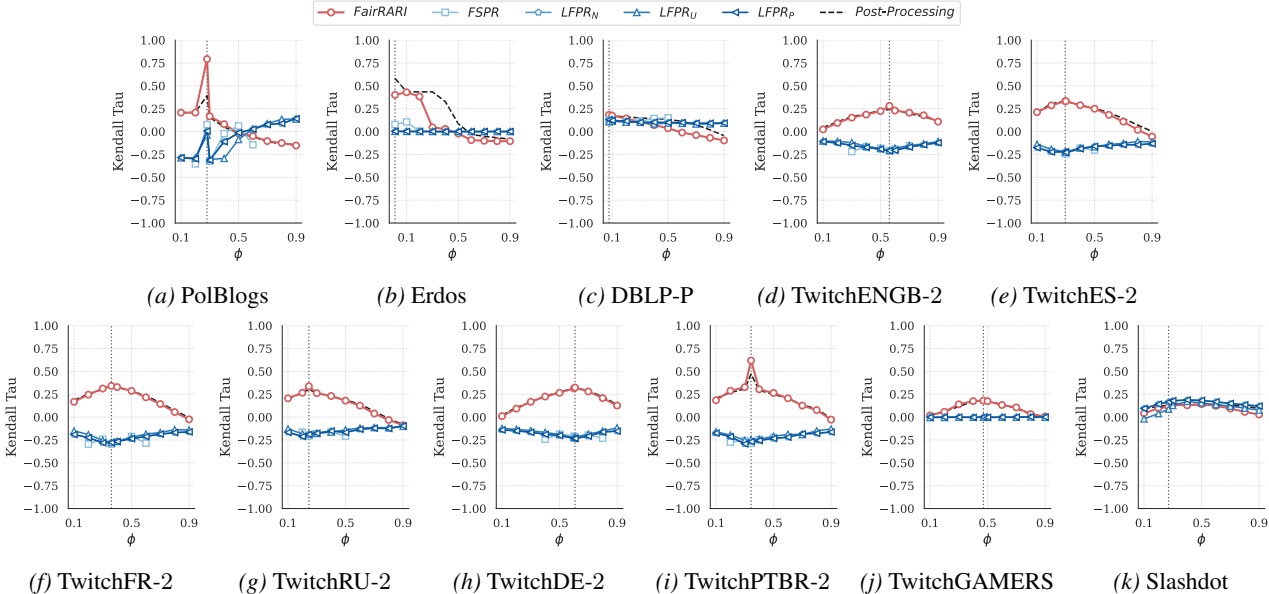

*Figure 10.* Extra experiments similar to Fig. 3. Comparing Fair PageRank solutions of prior methods and FairRARI (Ours), on the rest of datasets with 2 groups. It is evident that our algorithm outperforms the existing approaches. For various values of $\phi$, FairRARI consistently yields a significantly higher Kendall Tau values, indicating that the resulting ranking is much closer to that of the original PR.

### G.4.3. UTILITY LOSS

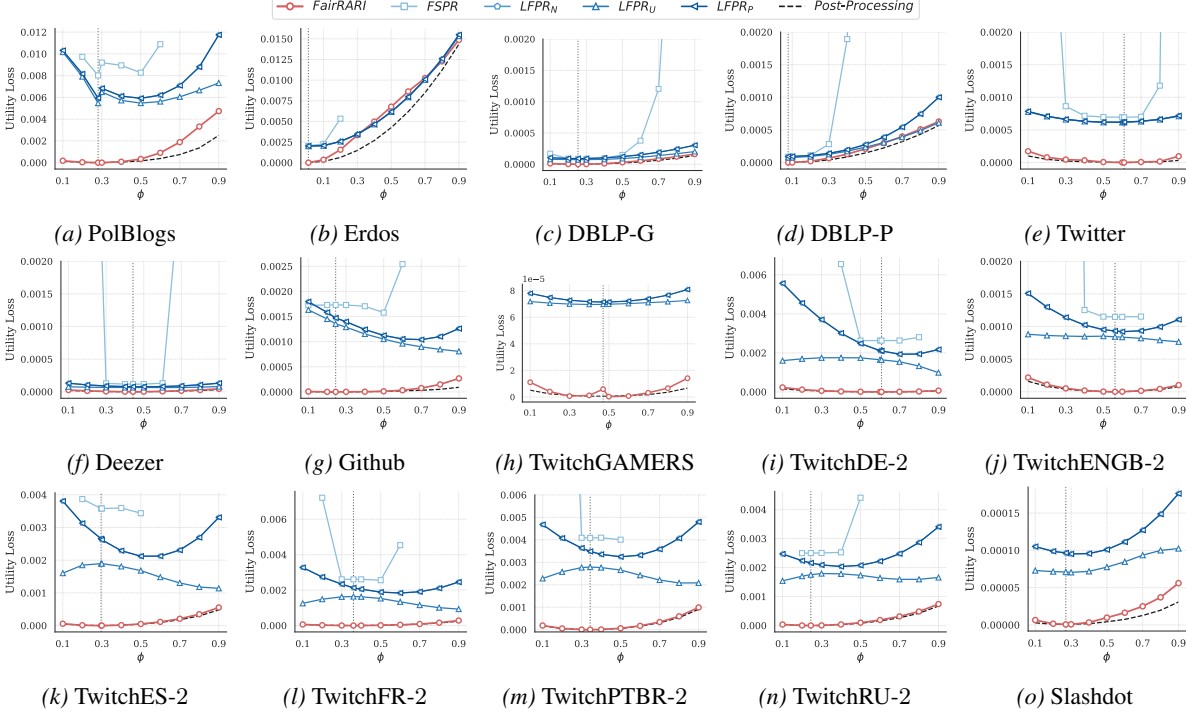

*Figure 11.* Utility Loss in terms of $\|\mathbf{p}_o - \mathbf{p}_F\|_2^2$, as defined in prior methods. The vertical dotted line represents the $\phi_o$ value of the original PR. It is evident that our algorithm outperforms the existing approaches in terms of this utility loss as well. For various values of $\phi$, FairRARI consistently yields a significantly lower loss, very close the lower bound obtained via the post-processing method, indicating that the allocation of fair PR scores is much closer to that of the original PR.

### G.4.4. $\phi$-PRECISION

To evaluate $\mathbf{p}_F$ based on how close the top-$x\%$ interval of the ranked vertices is to the desired $\phi$ value, we define the following measure, $\phi-\text{Precision@}x(\mathbf{p}_F) = (\mathbf{e}^\top_{\mathcal{T}_x \cap \mathcal{S}_k} \mathbf{p}_F)/(\mathbf{e}^\top_{\mathcal{T}_x} \mathbf{p}_F)$, where $\mathcal{T}_x$ represents the set of $x\%$ vertices with the highest PR scores. Hence, $\phi-\text{Precision@}x$ calculates the ratio of the scores of the vertices of a given group $\mathcal{S}_k$ in each top-$x\%$ interval. It can be interpreted as a weighted version of the Skew measure from (Geyik et al., 2019), where the weights are vertex scores. We adopt this measure because FairRARI and the baselines explicitly optimize the PR scores to provide group-fairness instead of only the ranking order.

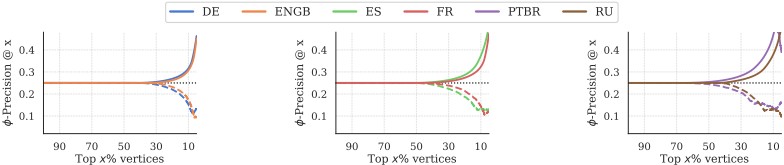

*Figure 12.* Showcasing $\phi$-Precision of prior methods and FairRARI across top-$x\%$ intervals. The horizontal dotted lines represent the target fairness value $\phi$. As $x\%$ increases, all methods attain $\phi$. However, the $\phi$-Precision of FairRARI is generally close to $\phi$ across varying $x\%$, demonstrating stable performance. In contrast, the prior methods often deviate sharply from the desired $\phi$ value, either overshooting or undershooting the target before attaining $\phi$, at a much larger $x\%$.

*Figure 13.* For the Twitch datasets with 4 groups, showcasing the min (dash-dot lines) and the max (dashed lines) value (across the different groups) of $\phi$-Precision for FairRARI across top-$x\%$ intervals. The horizontal dotted lines represent the target fairness value.

### G.4.5. FAIRNESS VIOLATION

In order to evaluate if a resulting solution vector $\mathbf{p}_F$ of a fair PR algorithm satisfies the $\phi$-sum-fairness constraints, we define the *fairness violation* $FV(\boldsymbol{\phi}_F, \boldsymbol{\phi}) := \frac{1}{K} \sum_{k=1}^{K} (\boldsymbol{\phi}_F(k) - \boldsymbol{\phi}(k))^2$, with $\boldsymbol{\phi}_F(k) = \mathbf{e}_{\mathcal{S}_k}^\top \mathbf{p}_F$ denoting the sum of the scores of the solution of fair PR, for each group. Furthermore, we denote as $\boldsymbol{\phi}_o$ the vector that represents the sum of the scores of each group in the original PR solution, i.e., $\boldsymbol{\phi}_o(k) = \mathbf{e}_{\mathcal{S}_k}^\top \mathbf{p}_o, \forall\, k \in [K]$.

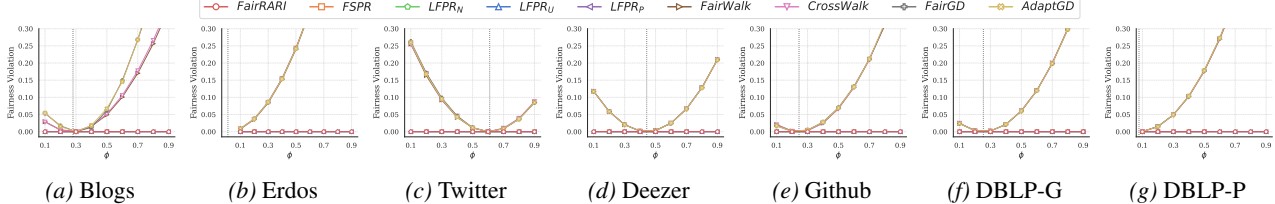

*(a)* Blogs     *(b)* Erdos     *(c)* Twitter     *(d)* Deezer     *(e)* Github     *(f)* DBLP-G     *(g)* DBLP-P

*Figure 14.* Comparing the Fairness Violation in the 2 groups case - making clear the unsuitability of FairWalk, CrossWalk, FairGD and AdaptGD in our setting. Except for the Blogs dataset, FairWalk, CrossWalk, FairGD, and AdaptGD produce the same fairness violation values on all remaining datasets.

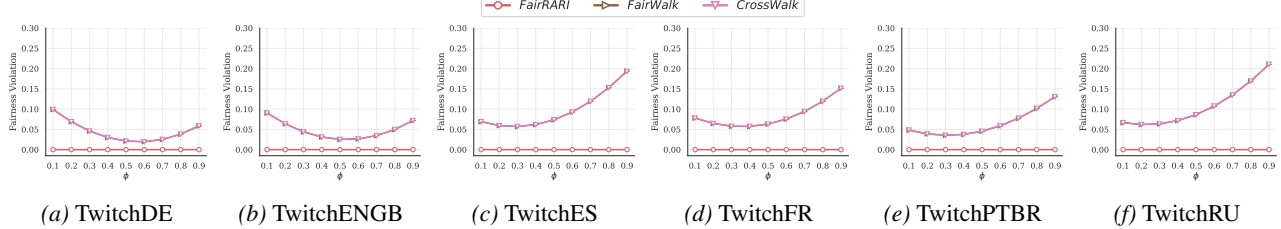

*(a)* TwitchDE     *(b)* TwitchENGB     *(c)* TwitchES     *(d)* TwitchFR     *(e)* TwitchPTBR     *(f)* TwitchRU

*Figure 15.* Comparing the Fairness Violation in the 4 groups case - making clear the unsuitability of FairWalk and CrossWalk, in the multiple groups case, as well.

### G.4.6. MULTIPLE GROUPS

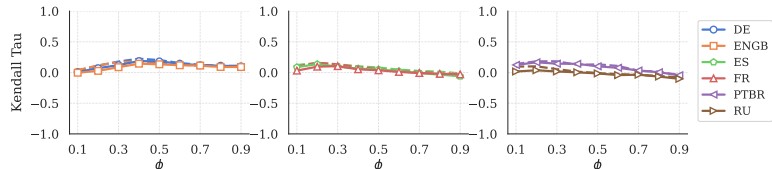

*Figure 16.* Showcasing the values of the Kendall Tau coefficient the FairRARI solutions for different fairness levels $\phi$, on Twitch datasets with 4 groups. The dashed lines represent the post-processing solution for each dataset.

### G.5. $\alpha$-min-fairness

For the following experiments we used only datasets with two groups. From the first group $\mathcal{S}_1$ we create $\mathcal{A} \equiv \mathcal{S}_1$ with $\alpha_1 = \alpha$, and for the second one $\alpha_2 = 0$. In order to have different target min-fairness levels, we create a vector of $\alpha$ values of the following form $\boldsymbol{\alpha} = [0.005, 0.01, \ldots, 0.05, 0.075, 0.1, 0.2, \ldots, 0.9]/n_1$, creating the x-axes of Fig. 17.

In Fig. 17 we present the TV attained for different $\alpha$ values. In this case, the vertical dotted line represents the minimal value, $\alpha_o$, that the original PR assigns in group $\mathcal{A}$. As expected, the TV below this $\alpha_o$ value is always 0, as the solution of FairRARI is the same as that of original PR, which already satisfies the min-fairness constraints for $\alpha \leq \alpha_o$. Moreover, we present the minimum score given to the vertices in group $\mathcal{A}$, for the different $\alpha$ values, showcasing that FairRARI method always satisfies the $\boldsymbol{\alpha}$-min-fairness constraints.

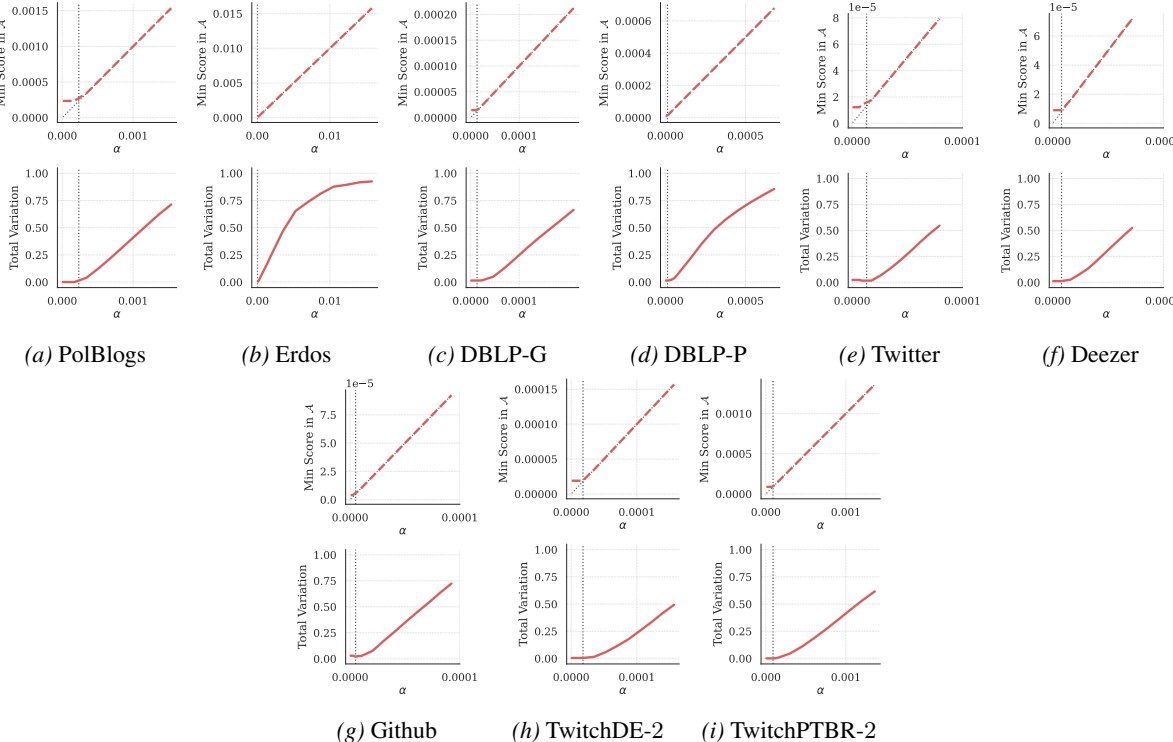

*Figure 17.* Showcasing the performance of the FairRARI solutions on the $\boldsymbol{\alpha}$-min-fair problem, on different datasets with 2 groups. Top: The minimum score assigned to the vertices in group $\mathcal{A}$, for different $\alpha$ values. Bottom: POF paid for different values $\alpha$. The diagonal dotted line represents the identity line and the vertical one the $\alpha_o$.

### G.6. $\phi$-sum + $\alpha$-min-fairness

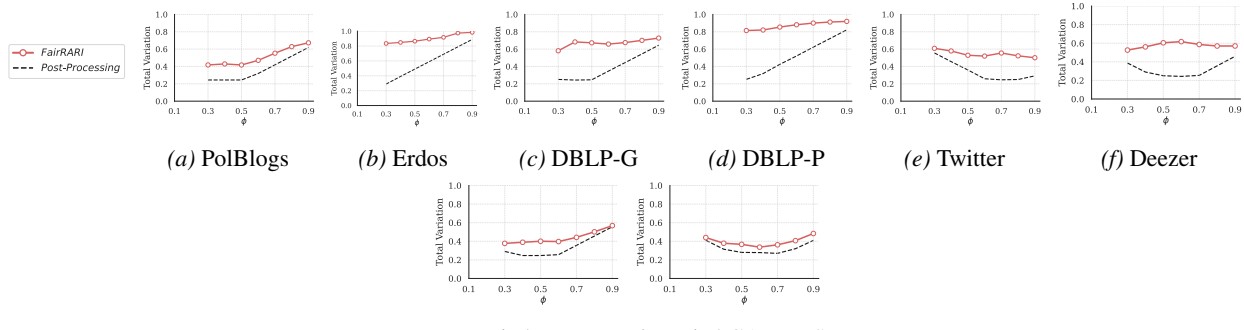

*(a)* PolBlogs    *(b)* Erdos    *(c)* DBLP-G    *(d)* DBLP-P    *(e)* Twitter    *(f)* Deezer

*(g)* TwitchPTBR-2   *(h)* TwitchGAMERS

*Figure 18.* Showcasing the performance, in terms of TV, of the FairRARI and Post-Processing solutions on the $\phi$-sum + $\alpha$-min-fair problem, on different datasets with 2 groups.

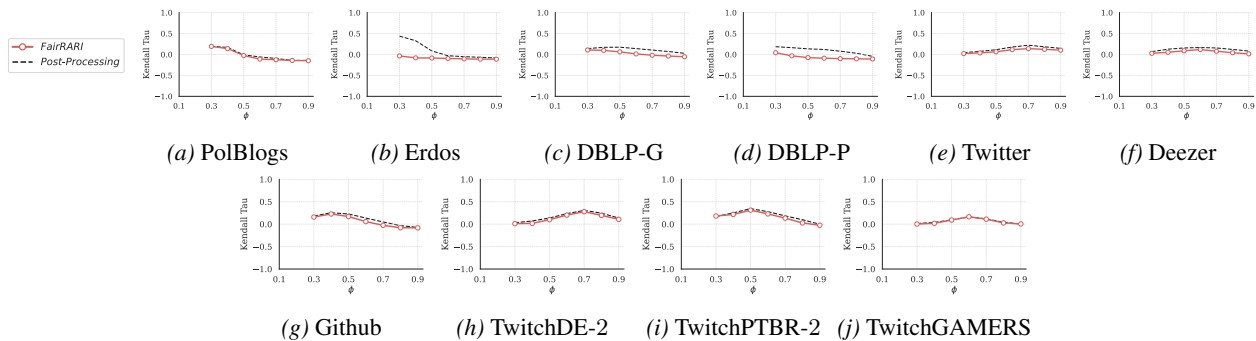

*(a)* PolBlogs    *(b)* Erdos    *(c)* DBLP-G    *(d)* DBLP-P    *(e)* Twitter    *(f)* Deezer

*(g)* Github    *(h)* TwitchDE-2    *(i)* TwitchPTBR-2   *(j)* TwitchGAMERS

*Figure 19.* Showcasing the performance, in terms of Kendall Tau coefficient, of the FairRARI and Post-Processing solutions on the $\phi$-sum + $\alpha$-min-fair problem, on different datasets with 2 groups.

