# OpenReview forum: "FairRARI: A Plug and Play Framework for Fairness-Aware PageRank"
_ICML.cc/2026/Conference — ICML 2026 regular_

### Official Review · Reviewer_6L5b · 2026-02-27

**Soundness:** 4
**Presentation:** 4
**Significance:** 4
**Originality:** 3
**Overall Recommendation:** 5
**Confidence:** 3

**Summary:**

This paper considers the task of ensuring fairness in the PageRank (PR) algorithm. To achieve this, they first formulate the fixed point vector for PR as the unique minimizer for a quadratic optimization problem for both undirected and directed graphs. Assuming that the fairness constraints can be expressed as a closed convex set, they propose the solution to finding a fair vector as a result of PR should be the one that minimizes the quadratic optimization problem, subject to the constraint that the vector is fair. To solve this problem, they propose the following modification of the iterations to find the fixed point of the PR algorithm: At each step, project the obtained vector onto the set of fair vectors. They show that this new iterative step, with the projection, still converges at a geometric rate to the fair vector that minimizes the quadratic optimization problem equivalent to PR. While the projection can be done by standard quadratic optimization solvers, the paper shows that for common notions of fairness - demographic parity, and lower bounds on the utility of the worst-off individual (Rawlsian social welfare) - that the projection can be done in much faster time by leveraging the structure of the constraint set. Finally, the authors perform extensive experiments to show that their method finds fair PR vectors that are much closer to the PR vector without fairness (in TV distance), suggesting they find much higher quality vectors, while guaranteeing fairness is achieved.

**Compliance With Llm Reviewing Policy:**

Affirmed.

**Final Justification:**

Overall, I think the paper is reasonably strong.

The problem definitely relevant for studying fairness, and the authors have provided a method to improve fairness of PageRank and provided theoretical and empirical guarantees to support this. This work has a key advantage over existing literature as this method guarantees the fairness constraints are met, which is not necessarily true in previous work. The authors have also provided interesting insights through their analysis, such as in their discussion on why a single postprocessing step (find closest fair vector after solving PageRank) is less ideal compared to applying the fairness step at each iteration. The authors have also studied their approach empirically on many real-world datasets which validate the approach.

While there may be some weaknesses - for instance, there may be fairness constraints that cannot be represented as convex sets - these do not detract too much from the strengths as the authors have studied and proven that their approach works for well-studied and used fairness metrics.

Given these factors, I believe the work is sound theoretically and empirically, and is definitely significant with potential for others to build upon the ideas or to apply the approach. The conceptual idea of postprocessing to find a closest fair vector is similar to approaches such as in fair clustering and fair ranking, but the setting of PageRank requires totally different analysis, therefore I believe there is sufficient novelty. The paper is written well and I have no complaints on the structure or clarity, and the contributions are clear.

Their rebuttal has also addressed my main concerns. While the theoretical assumptions on a directed graph are somewhat strong, the approach can still be applied in practice.

**Key Questions For Authors:**

1. I suppose that the $\phi$-sum-fairness be relaxed to account, for example, a fairness constraint of the form that for each group, the sum of PR scores is lower bounded by some $\alpha$ and upper bounded by some $\beta$? I am curious if there were fairness notions that you encountered relevant to PageRank that were not able to be expressed as a closed convex set?

2. In the datasets with directed edges, did the markov chains satisfy time-reversibility?

**Limitations:**

yes

**Strengths And Weaknesses:**

I found this paper quite interesting. My understanding is that the author's formulation of fairness in PR can be seen as an idea analogous to finding 'closest fair cluster' or 'closest fair ranking' in works that study algorithmic fairness in clustering/rankings.

Strengths:

All theorems and lemmas are supported by theoretical analysis. Empirical results also show their method finds a fair PR vector much closer to the (unfair) PR vector than previous approaches as well. Claims on achieving efficient running time are supported as well theoretically and empirically. The experiments also encompass a large number of real-world networks to provide realistic empirical analysis.

The fairness constraints that are considered are well supported by existing work in algorithmic fairness. The approach proposed also guarantees the fairness constraints are met, compared to previous approaches which only do so for 2 groups.

I think that improving fairness PageRank is definitely significant, especially given it can have applications in areas like social media reach, recommendation systems and online search; all areas where ensuring fairness in the PR vector scores for different groups to help mitigate biases.

The paper is written well and I had no issues following it. It is nice the authors also gave some intuition e.g. for (4) and section 5.

Weaknesses:

Maybe the one thing I notice is the assumption of time-reversibility for the random walk on directed graphs, which likely would not hold in practice in networks with directed links (like follows in a social media network).

---

> ### Author Rebuttal · Authors · 2026-03-27
>
> We thank the reviewer for their positive evaluation of our paper. The comments are carefully addressed below.
>
> Weakness:
> Please see our response to W1 and W2 of Reviewer S5Jk.
>
> Questions:
>
> Q1:
> Indeed, for 2 groups, the relaxed $\phi$-sum-fairness constraint $\alpha \leq e^T_{S_0}x\leq \beta$ still defines a closed and convex set that can be handled directly within the FairRARI framework. The extension to more than two groups follows analogously; however, selecting the upper and lower bounds for each group becomes more delicate, as they must be chosen carefully to ensure the overall constraint set remains feasible.
>
> We thank the reviewer for this insightful question.
> While many fairness criteria in PR can be naturally expressed using closed convex sets (Section 4), non-convex constraints can also arise. We have observed that the sum-fairness criterion can lead to disparities in allocation of PR scores within a group (see Fig. 1). To rectify this, we can impose an additional per-group constraint that controls the variance of the allocated PR scores within a group. These can be expressed as  $\alpha_k \leq (1/n_k)\cdot\sum_{i \in \mathcal{S}_k} (x_i - (\phi_k/n_k))^2\leq \beta_k$. Here, $(\phi_k/n_k)$ is the average PR score of a node in group $\mathcal{S}_k$ comprising $n_k$ nodes and target PR mass $\phi_k$. The upper and lower bounds ensure that the PR scores do not stray too far or stay too close to the mean. Clearly, these constraints are non-convex. Since tackling such constraints is a difficult proposition, we used the min-fairness criterion (which is convex) to address these imbalances.
>
> Q2:
> As per our response to W1 of Reviewer S5Jk, we applied the time-reversibility test on the random walk matrix $P$ of the directed graphs used in our study. In the leftmost column of the tables below, we display the Frobenius norm error, $\text{Error} := \lVert \Pi P^T - P \Pi \rVert_F$, and the Mean Absolute Error (MAE) in the balance equations (30). Although these results indicate that time-reversibility does not hold *exactly*, it is evident that $P$ well approximates this property. We additionally tested for time-reversibility of the combined Markov chain $\tilde{P} := (1-\gamma)P + \gamma (1/n)ee^T$ for increasing values of $\gamma$, and we depict the error metrics in the remaining columns of the tables below. The observed decrease in both metrics as $\gamma$ increases provides empirical confirmation that incorporating teleportation drives $\tilde{P}$ toward time-reversibility. At the limit of $\gamma = 1, \tilde{P}= (1/n)e e^T$, which corresponds to a fully time-reversible Markov chain.
>
> | | PolBlogs ($\gamma=0$) | PolBlogs ($\gamma=0.15$) | PolBlogs ($\gamma=0.85$) |
> | :--- | :---: | :---: | :---: |
> | Error | $0.0183$ | $0.0118$ | $0.0019$ |
> | MAE | $10\cdot 10^{-7}$ | $9\cdot 10^{-7}$ | $2\cdot 10^{-7}$ |
>
> | | Slashdot ($\gamma=0$) | Slashdot ($\gamma=0.15$) | Slashdot ($\gamma=0.85$) |
> | :--- | :---: | :---: | :---: |
> | Error | $0.0007$ | $0.0006$ | $0.0002$ |
> | MAE | $6\cdot 10^{-11}$ | $4\cdot 10^{-11}$ | $3\cdot 10^{-11}$ |

---

> > ### Author Rebuttal · Reviewer_6L5b · 2026-04-01
> >
> > The authors have satisfactorily responded to the questions about the general applicability of some other natural fairness constraints to convex sets, and the assumptions on time-reversibility.

---

### Official Review · Reviewer_1ciE · 2026-03-10

**Soundness:** 3
**Presentation:** 3
**Significance:** 3
**Originality:** 3
**Overall Recommendation:** 4
**Confidence:** 3

**Summary:**

The authors study an area where algorithmic fairness intersects with fundamental graph machine learning primitives, specifically addressing the lack of principled fairness-aware algorithms for PageRank. This paper mainly focus on putting forth a unified in-processing convex optimization framework, termed FairRARI, for tackling different group-fairness criteria in a"plug and play"fashion. Key contributions include a variational formulation of PageRank that enables principled fairness constraints, provable convergence to optimal fair solutions, and linear-time projections that maintain the asymptotic complexity of standard PageRank. Extensive experiments demonstrate that FairRARI outperforms existing baselines in utility while strictly satisfying target fairness levels across multiple vertex groups.

**Compliance With Llm Reviewing Policy:**

Affirmed.

**Key Questions For Authors:**

see weakness

**Limitations:**

yes

**Strengths And Weaknesses:**

### strengths
* The authors propose FairRARI, a unified in-processing convex optimization framework that handles multiple group-fairness criteria in a "plug and play" fashion with provable guarantees.
* The method converges to the unique optimal fair solution via simple fixed-point iterations while maintaining the same asymptotic time complexity as standard PageRank through linear-time projections.
* Extensive experiments on 22 real-world datasets demonstrate that FairRARI consistently outperforms existing baselines in utility (TV distance and Kendall Tau) while strictly satisfying target fairness levels across multiple vertex groups.
### weaknesses
* The experimental section lacks performance evaluation on downstream machine learning tasks (e.g., node classification, link prediction). While the paper evaluates utility based on TV distance and Kendall Tau correlation with the original PR vector, it does not demonstrate whether the fair PR vectors produced by FairRARI actually improve fairness-utility trade-offs in practical applications.
* Although the paper proves that FairRARI shares the same asymptotic time complexity as standard PageRank, each iteration requires executing a projection operation (involving binary search to solve for dual variables). Experiments show approximately 10% increase in runtime overhead. However, this constant-factor overhead should be considered in ultra-large-scale graphs or scenarios with stringent real-time requirements. Furthermore, the complexity of parallel implementation is not sufficiently demonstrated in the main experiments.

---

> ### Author Rebuttal · Authors · 2026-03-27
>
> We thank the reviewer for their positive evaluation of our paper. The comments are carefully addressed below.
>
> Weaknesses:
>
> W1: In this paper, our goal is to promote group fairness in the allocation of PageRank scores from the perspective of node centrality, which is the original objective of the classic PageRank algorithm. This choice guided both the design of our fairness constraints and selection of evaluation measures - specifically, TV distance and Kendall Tau correlation, which directly reflect centrality-based utility. The FairRARI framework can also be generalized to accommodate fairness notions that are relevant for other downstream tasks of PR such as node classification and link prediction. The key observation is that the variational form of the non-fair PR problem derived herein enables incorporation of various fairness criteria tailored to specific downstream tasks. We acknowledge the development of such extensions as valuable directions for future research, which we shall mention in the revised version.
>
> W2: In our experiments, we used the sequential implementation of the projections in FairRARI for the considered datasets. The increased runtime overhead of approximately $10\%$ is modest and does not constitute a significant bottleneck in these settings. For large-scale graphs, where such overheads can prove significant, the parallel implementation of the projection step is a natural remedy. In the revised version, we will add the runtimes on the largest dataset TwitchGamers with the parallel implementation to demonstrate the speed-ups over its sequential counterpart (Figures 6(a), 7(a) and 8(a)),
> complementing the existing theoretical complexity analysis. Please also see our response to W3 of Reviewer S5Jk.

---

> > ### Author Rebuttal · Reviewer_1ciE · 2026-04-03
> >
> > N/A

---

### Official Review · Reviewer_S5Jk · 2026-03-14

**Soundness:** 3
**Presentation:** 3
**Significance:** 2
**Originality:** 2
**Overall Recommendation:** 4
**Confidence:** 4

**Summary:**

The authors propose FairRARI for the fair personalized PageRank problem. They formulate the ranking problem as a regularized optimization objective (a quadratic convex objective). To improve the problem group's fairness, they add a fairness-aware correction step that preserves ranking quality. The novelty part of FairRAPI is to formulate the problem from a variational perspective. They provide theoretical guarantees for the optimization formulation and convergence behavior. The experimental results show that the approach can improve fairness-utility tradeoffs compared with baseline PageRank-style methods.

**Compliance With Llm Reviewing Policy:**

Affirmed.

**Key Questions For Authors:**

Overall, I think this is a decent paper. I have the following questions:

1. How can one identity the source-directed graph is reversible or not? This is the key to applying your proposed method.

2. Please justify the scalability of the proposed method when the underlying graphs are huge-scale.

3. A question about the fairness vector: The paper sets the target fairness vector for multiple groups as $\phi = (\phi, 1 − \phi)$. It gives the first group $\phi$ and distributes the remaining mass $1-\phi$ equally among the other groups. Could the authors explain the motivation for this design? In particular, this choice ignores differences in group sizes and implicitly treats the first group differently from the others.

**Limitations:**

yes

**Strengths And Weaknesses:**

Overall, the authors study an interesting research problem of PageRank concerning the fairness. The proposed algorithm has some nice theoretical guarantees and the experimental results verify the proposed method. I list strong and weak points as follows:

Strong points:

- 1. The authors study an important problem about the fairness of PageRank, which keeps the method practical and compatible with existing PageRank pipelines.

- 2. The FairRARI method is conceptually simple and modular: the “plug-and-play” design makes the approach easy to understand and potentially easy to adopt in existing approaches.

- 3. The authors developed theoretical results and had empirical evaluation, which helps connect the method’s intuition with its practical behavior.

Weak points:

- 1. PageRank mainly focuses on directed graphs. However, some theoretical assumptions seem strong, especially the reversibility assumption used for the directed-graph formulation. This may substantially limit the claimed generality for directed graphs. The authors may justify how to identify whether the real-world directed graph is reversible or not.

- 2. The paper should more clearly distinguish what is guaranteed for general directed graphs versus only for reversible directed chains, since this affects how broadly the method applies.

- 3. The datasets used in this paper (listed in Table 1) are quite small graphs. It is unclear whether the proposed algorithm is scalable to large-scale graphs, which are quite common in modern graph learning.

---

> ### Author Rebuttal · Authors · 2026-03-27
>
> We thank the reviewer for their positive evaluation of our paper. The comments are carefully addressed below.
>
> Weak points:
>
> W1 + Q1: Given a  directed graph $G$, testing for time reversibility boils down to checking the balance equations (29)-(30). A necessary condition is that if a directed arc from node $i$ to $j$ exists, a "reverse" arc from $j$ to $i$ must also exist. If this test fails for any node pair, we can conclude that time-reversibility does not hold. If this test passes, we can then proceed to check if the balance equation (30) is satisfied. This can be done by computing the stationary distribution $P\pi = \pi$ of the random walk matrix $P$, setting $\Pi = diag(\pi)$, and then checking whether $\Pi P^T = P\Pi$ is satisfied or not.
>
> We acknowledge that such an assumption on $P$ is strong. It can be weakened to the condition that the combined Markov chain with transition matrix
> $\tilde{P} := (1-\gamma)P + \gamma ve^T$, $v=(1/n)e^T$ (since we measure centrality) is time-reversible. This is a more justifiable assumption, as the incorporation of teleportation ensures that the necessary condition for balance is satisfied; i.e., $\tilde{P_{ij}} > 0 \Leftrightarrow  \tilde{P_{ji}} > 0$. Moreover, the transition matrix $(1/n)\cdot ee^T$ is time-reversible by design. Hence, as $\gamma \rightarrow 1$, $\tilde{P}$ better approximates a time-reversible Markov chain. We performed the time-reversibility test for $\tilde{P}$ on the directed graph datasets that we used, and discovered that they very closely approximate this property even for $\gamma = 0.15$ ($1\\%$ Frobenius norm error for PolBlogs and $0.06\\%$ for Slashdot);  please see our response to Q2 of Reviewer 6L5b.
>
> In this setting, we can establish a different variational form of non-fair PR
> as minimizing the quadratic function $\tilde{f}(x) := \frac{1}{2}x^T \Pi^{-1}\tilde{L_{rw}} x,$ where $\tilde{L_{rw}} := I - \tilde{P}$. By employing the same arguments as in the proof of Theorem B.1 (App B.1), we can show that time-reversibility of $\tilde{P}$ implies that
> $\Pi^{-1}\tilde{L_{rw}}$ is positive definite, and hence, $\tilde{f}(x)$ is strongly convex. Furthermore, we can directly generalize the proof of Theorem 3.3 to assert that the unique minimizer of $\min_{x\in \mathcal{X}} \tilde{f}(x)$ coincides with the unique fixed point of the FairRARI iterations. This follows from our established proof technique in Appendix B.3 by noting that: $\Pi \nabla \tilde{f}(x) = [I-(1-\gamma)P]x - \gamma v$, and repeating the remaining arguments verbatim. We will add this more general result to the revised version, as it follows directly from the proofs of Theorems B.1 and 3.3.
>
> W2: We agree that the scope of our theoretical results warrants clearer exposition. The assumption of time-reversibility is only required to establish Theorem 3.3. In contrast, Theorem 3.2 holds for *every* directed graph; i.e., the FairRARI iterations always converge to a unique fixed point, regardless of whether time-reversibility is satisfied. This implies that the algorithm is broadly applicable in practice, but interpreting its fixed point as the solution of a fair PageRank optimization problem requires an extension of Theorem 3.3 to non-reversible settings. The main technical challenge is the absence of a canonical graph Laplacian in this setting (as opposed to undirected graphs) which would facilitate a variational interpretation of PR. We will explicitly clarify the scope of each result in the revised manuscript.
>
> W3 + Q2: The datasets used in our experiments include the largest currently available with sensitive nodal attributes needed for group-fairness evaluation. They represent a substantial increase in scale compared to those used in the prior art on fair PageRank (see Section 6 - Baselines) - most notably TwitchGamers - and align with the dataset sizes commonly used in the fair graph learning literature [1, 2, 3]. Beyond empirical evaluation, we provide a rigorous proof that FairRARI achieves the same asymptotic time complexity as the original PageRank algorithm (page 6, before Section 5), offering a theoretical foundation for its scalability. Please also see our response to W2 of reviewer 1ciE.
>
> [1] https://doi.org/10.48550/arXiv.2102.13186 \
> [2] https://doi.org/10.1145/3534678.3539404 \
> [3] https://doi.org/10.1145/3485447.3512173
>
>
> Questions:
>
> Q3: We chose to parameterize the target fairness vector using a single variable $\phi$ to provide a clear and interpretable visualization of the fairness-utility trade-off (Fig. 4). Although this specific setup treats one group differently from the others, it serves as an illustrative  proof of concept. This choice does not reflect any technical limitation of the framework, which is guaranteed to fulfill any choice of target fairness levels $\\{\phi_k\\}_{k=1}^{K}$ across multiple groups.

---

> > ### Author Rebuttal · Reviewer_S5Jk · 2026-04-04
> >
> > Thank you for these responses. I will keep my score unchanged.

---

### Decision · Program_Chairs · 2026-04-30

**Decision:**

Accept (regular)

**Comment:**

The work gives principled algorithms that generalize the PageRank algorithm while guaranteeing group fairness conditions via convex optimization. The reviewers appreciated the importance of the problem, the achieved theoretical guarantees, the conceptual simplicity and modularity, and the extensive empirical evaluation. The runtime is almost the same as that of the original PageRank algorithm (also in practice). The sentiment was positive across the board.

Some concerns were raised regarding the strong time-reversibility assumption that is necessary for part of the results. This has been addressed in the rebuttal to some degree; the authors are strongly requested to introduce their proposed improvements to the manuscript.